# Fluorescence-based mapping of condensate dielectric permittivity uncovers hydrophobicity-driven membrane interactions

E. Sabri [1], A. Mangiarotti [1,2,3] & R. Dimova [1] ✉

Biomolecular condensates, essential for cellular organization, possess mesoscale properties largely governed by hydrophobicity, influencing molecule partitioning and material characteristics like viscosity, surface tension, and hydration. While hydrophobicity's role is increasingly recognized, its impact on membrane-condensate interactions remains unexplored. Here, we combine hyperspectral imaging of an environment-sensitive dye and phasor analysis, to quantitatively map the local dielectric permittivity of both condensates and their environment with pixel resolution. This robust method senses the immediate molecular vicinity of the dye and reveals a surprisingly broad range of condensate permittivities, spanning from oil-like to water-like values. Importantly, we uncover that membrane affinity is not dictated by condensate permittivity itself, but by the permittivity contrast with their surroundings. Indeed, membrane wetting affinity is found to scale linearly with this permittivity contrast, unveiling a unifying dielectric principle governing condensate-membrane interactions. Compatible with live-cell and in vitro imaging, this technique provides quantitative insights into condensate biophysics and function and opens new avenues for studying biomolecular condensate biology.

Biomolecular condensates form as a result of liquid-liquid phase separation (LLPS) and are key to intracellular organization. Examples of these phase-separated compartments include membraneless organelles such as stress granules[1], nucleoli[2], mitochondrial nucleoids[3], and nuclear speckles[4], which provide a dynamic and reversible means for cells to maintain homeostasis[1,3,4], adapt to stress[5–7], and regulate biochemical processes spatially and temporally[4,8,9]. The unique biophysical properties of condensates, such as viscosity[10–12], protein packing[13,14], hydration[7,13,14], pH[15], surface tension[11], and electrostatic properties[16], set them apart from their surrounding environment and are crucial for their diverse functions. In particular, the hydrophobicity and perceived micropolarity of condensates have recently emerged as readily measurable characteristics exhibiting strong correlations with each of these physicochemical parameters[9,11,17,18].

While terms like "polarity"[19,20] or "micropolarity"[11,18,21] are commonly encountered, their use can be misleading when referring to a material's dielectric constant, or dielectric permittivity. Dielectric permittivity, a macroscopic-scale property, reflects not only the polarity of molecules (i.e., the degree of charge separation at the molecular scale) but critically, also their capacity to reorient in the presence of a transient electric field (e.g., originating from the transiently excited state of a dye dipole). A molecule can possess high intrinsic polarity (i.e., a pronounced charge separation), but if its rotational mobility is restricted, its contribution to the overall

[1]Max Planck Institute of Colloids and Interfaces, Science Park Golm, Potsdam, Germany. [2]Centro de Investigaciones en Química Biológica de Córdoba (CIQUIBIC), CONICET, Córdoba, Argentina. [3]Departamento de Química Biológica Ranwel Caputto, Facultad de Ciencias Químicas, Universidad Nacional de Córdoba, Córdoba, Argentina. ✉e-mail: Rumiana.Dimova@mpikg.mpg.de

dielectric permittivity of the medium will be low, despite its high molecular polarity. Provided that the dipolar relaxation of water molecules is the main factor of molecular reorientation in biological samples[22], we will primarily use the more suitable terms "hydrophobicity" or "dielectric permittivity". To ensure broad accessibility and to align with existing terminology in certain contexts, we will also employ "micropolarity" where appropriate, particularly when referencing established measurement paradigms.

Among the various techniques used to characterize condensates, fluorescence-based imaging has emerged as a powerful tool due to its sensitivity, non-invasiveness, and high spatial and temporal resolution[11,17,23–26]. From a fundamental perspective, fluorescence-based approaches quantify micropolarity on a dielectric permittivity scale, relying on the degree to which water molecules can redshift or quench the emission of environment-sensitive dyes[11,18–21]. Notably, these permittivity-sensitive dyes can detect how changes in protein sequence and interactions influence condensate properties in vitro and in vivo[11,14,27]. Given these insights, it is reasonable to predict that the dielectric permittivity of condensates is a key determinant in modulating their interactions with surrounding cellular components, such as membrane-bound organelles and the cytoskeleton. However, this critical aspect remains largely unexplored, primarily due to technical challenges associated with quantifying micropolarity and establishing reliable calibration curves[11,17]. Furthermore, while many environment-sensitive fluorescent probes exist[11,17–19,21,28], their detection range is limited, failing to assess the sharp contrasts in dielectric properties, which are hallmark features, endogenous to biological and biomimetic systems.

In this study, we introduce a robust, fluorescence-based method for quantifying the dielectric permittivity of biomolecular condensates and their surrounding environment. Our approach utilizes 2-acetyl-6-(dimethylamino)naphthalene (ACDAN), a dipolar relaxation-sensitive dye[29,30], widely used to probe intracellular water activity[31–33], macromolecular crowding[22], and membranous compartments[30,34]. As with any solvatochromic probe, ACDAN reports the dielectric properties of its immediate molecular environment. While this dye has previously been employed to infer dipolar relaxation in biomolecular condensates, these applications relied on inherently qualitative, arbitrary unitless scales. Here, we overcome this limitation by proposing a rigorous framework to establish a quantitative and broadband estimation of condensate dielectric permittivity based on the spectral response of ACDAN. Importantly, our method enables a simultaneous measurement of the properties of both dense (condensate) and depleted (environment) phases.

Building on this, we investigated the hypothesis that condensates dielectric properties govern their interaction with cellular structures, particularly membranes. While prior work has shown that condensate-membrane affinity can depend on bilayer and protein hydrophobicity, even with unchanged membrane surface composition[35,36], the direct role of condensate dielectric properties in these interactions remained unclear. Our results demonstrate a clear correlation between membrane binding and the permittivity contrast between condensed and depleted phases, revealing that dielectric mismatch directly governs condensate-membrane affinity. These findings establish the permittivities of the coexisting phases as key quantifiable parameters for understanding how biomolecular condensates interact with diverse cellular structures such as membrane-bound organelles and—via modulation of surface adhesion energies—potentially other components such as the cytoskeleton.

## Results

### ACDAN hyperspectral imaging provides a highly sensitive readout of solvent permittivity

The fluorescence of the ACDAN dye (Fig. 1a) is highly sensitive to solvent dipolar relaxation[22,30–34]. Upon UV absorption, the dipole

moment of ACDAN increases, generating a local electric field. Surrounding solvent molecules reorient to align with this field, lowering the energy of the dye-solvent system. This solvent reorganization leads to a redshift of the fluorescence emission, with larger shifts corresponding to environments of higher dielectric permittivity (Fig. 1b). In low permittivity solvents, weak dipolar relaxation results in minimal redshift of the dye's fluorescence. Conversely, highly polar solvents lead to strong redshift due to enhanced solvent relaxation and a greater decrease in the energy gap for emission.

To translate this solvent-dependent spectral redshift into a quantitative measure of dielectric permittivity (i.e., build a calibration curve), we characterized ACDAN's spectral response in a broad panel of homogeneous solvents with known permittivity, using both hyperspectral microscopy (Fig. 1c, d) and spectrofluorimetry (see SI). While both approaches yield equivalent permittivity calibrations for homogeneous samples (Figs. S1–S3), spectrofluorimetry is limited to bulk (cuvette) measurements and lacks spatial resolution, limiting its ability to resolve distinct dielectric properties in phase-separated (or heterogeneous) samples. We therefore employ hyperspectral microscopy, which records a full emission spectrum at each image pixel, generating spatially resolved spectral maps (Fig. 1c, d). This allows local permittivity variations to be visualized directly within heterogeneous systems, including phase-separated biomolecular condensates.

To construct a calibration curve, we measured homogeneous reference solutions of known permittivity (Fig. 1d) and analyzed the spectra using two approaches: Gaussian fitting and a fit-free Fourier-space analysis based on spectral phasors[22]. Both methods yield equivalent results for simple spectra, with the phasor approach offering additional robustness and extensibility for complex spectral profiles.

For the Gaussian fitting approach, spectral data from each solution were averaged across all pixels to obtain a high-fidelity emission curve (Fig. 1d), which was then fitted with a Gaussian function (Fig. 1c sketch and "Methods"). Figure 1e (orange data) presents the resulting calibration curve, linking ACDAN's maximum emission wavelength to solvent permittivity.

The phasor plot analysis involves computing the discretized Fourier transform of the dye emission spectrum and representing it as a single point in a two-dimensional "phasor plot" of the real (G) and imaginary (S) components (Fig. 1c, bottom right). The plot's modulus (distance from the center) provides information on the broadness of the spectrum, which can be affected by, e.g., fluorescence quenching[37], while its phase (angular position) reflects the emission spectrum center of mass, which directly depends on the permittivity of fluorophore microenvironment[22] (Fig. 1c bottom right). In summary, the phasor plot analysis provides a fit-free representation of fluorescence spectra, allowing each pixel to be mapped onto a two-dimensional space where spectral shifts, rather than absolute intensity, can be directly visualized and compared. In this context, the phasor position serves as a compact descriptor of the local dielectric environment sensed by ACDAN.

For a hyperspectral image stack (Fig. 1c top left), the set of spectra measured at individual pixels yields a scatter plot (cloud) in the phasor space (as sketched in Fig. 1c bottom right). To map permittivity in a given system, we built a calibration curve by analyzing images of homogeneous solutions with known permittivity. This generated a reference phasor plot (Fig. S2), from which the phase was used to build the calibration curve Fig. 1f (orange data).

As a proof of principle, Fig. 1e, f includes a dataset for ethanol–water mixtures (blue data points), with dielectric constants estimated using the celebrated Maxwell–Garnett law of mixtures[38] (see Table S3 and Fig. S1c for spectra). These data serve as an additional validation that in these systems, the ACDAN redshift reflects solvent permittivity rather than specific chemical interactions. We demonstrate in the SI that this approach yields equivalent results via

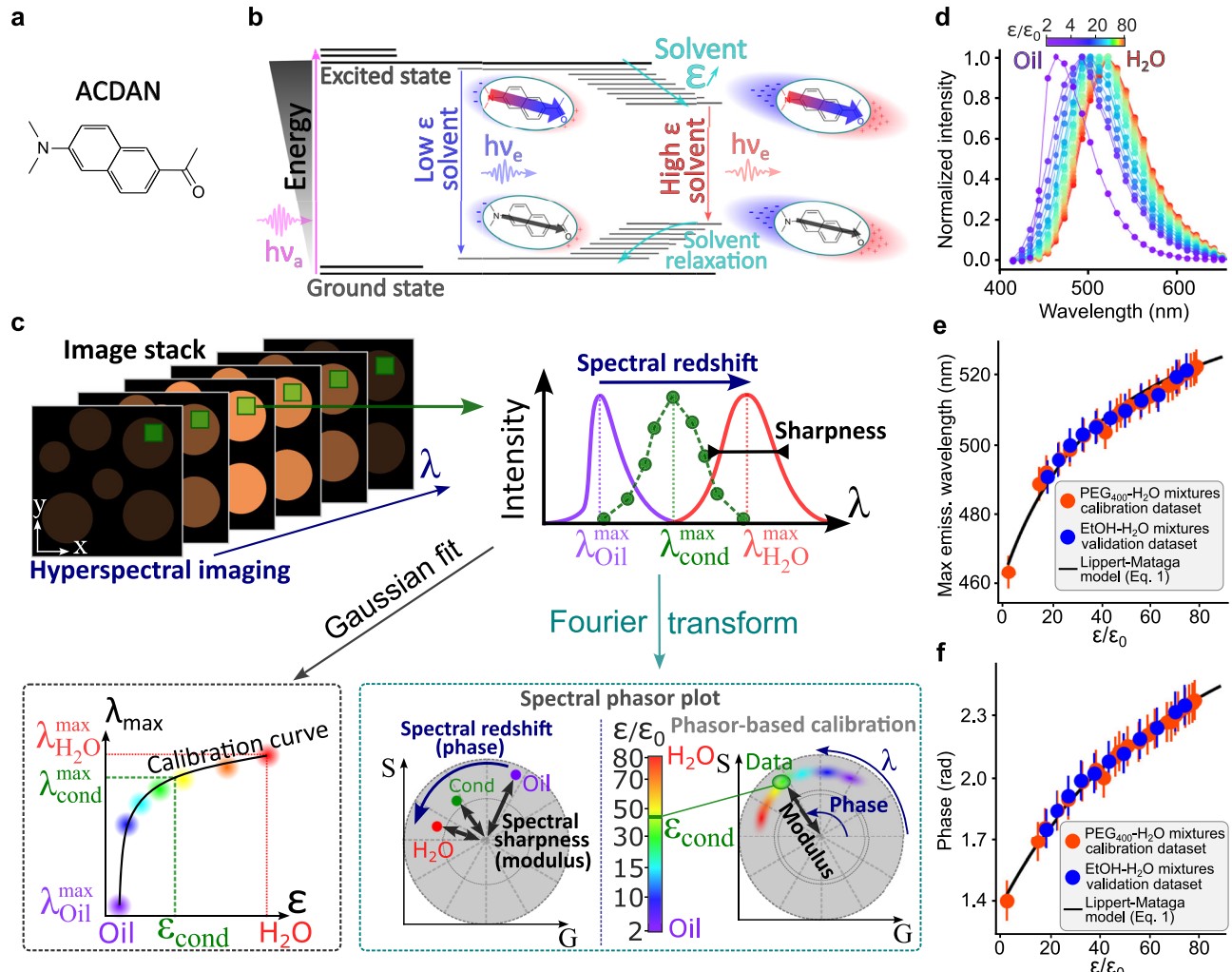

**Fig. 1 | ACDAN hyperspectral imaging enables spatial mapping of dielectric permittivity with pixel resolution. a** Molecular structure of the ACDAN (6-acetyl-2-dimethylaminonaphthalene) fluorescent probe, a water-soluble molecule that partitions into biomolecular condensates. **b** Perrin–Jablonsky diagram illustrating the dependence of ACDAN fluorescence on solvent permittivity. The sketch depicts changes in ACDAN dipole moment due to photon absorption or emission and the solvent relaxation process. $h\nu_a$ and $h\nu_e$ represent the absorption and emission energies, respectively. **c** Schematic representation of confocal hyperspectral imaging and analysis. A full emission spectrum is measured at each pixel from a stack of images acquired at different emission wavelengths ($\lambda$). The pixel permittivity (for either condensed or depleted phases) can be determined using two approaches: Gaussian fitting (bottom left) and phasor representation (bottom right), both detailed in the text. **d** Fluorescence emission profiles of ACDAN from hyperspectral microscopy imaging of calibration solutions of mineral oil or PEG400-H$_2$O solutions (from left to right: 100%, 90%, 80%, 70%, 60%, 50%, 45%, 40%, 35%, 30%, 25%,

20%, 15%, 10%, 8%, 6%, 4%, 2%, 1%, 0.5%, 0% PEG content). **e, f** Calibration data using the maximum emission wavelength of ACDAN obtained from a Gaussian fit of the spectrum (**e**) or the phase in the phasor plot (**f**) as a function of previously reported or theoretically calculated solvent dielectric constant ($\varepsilon$): Orange circles represent different PEG-400 solutions in water, with permittivity values from literature (referenced in Table S1). Blue circles represent ethanol-water mixtures, with dielectric constants calculated using the Maxwell–Garnett law of mixtures (see SI, Table S3 and Fig. S4b). The error bars indicate the spectral bandwidth used for detection, and datapoints represent the maximum emission wavelength averaged over all 512 × 512 pixels in the image stack ($n = 262144$). The relation $\frac{\phi(\lambda_f - \lambda_0)}{2\pi} + \lambda_0 = \lambda_{max}$ with $\lambda_0 = 416$ nm and $\lambda_f = 728$ nm was used to establish the link between the emission wavelength $\lambda_{max}$, and the phase $\phi$ for which the Lippert–Mataga equation (Eq. (1), solid curve; fitting parameters given in Table S2) accurately describes the data.

spectrofluorimetry (Figs. S1–S3), and that a fit-free phasor representation in Fourier space also yields equivalent results to the Gaussian fit (see Figs. S1–S3).

While both approaches yield equivalent results, this comparison serves several purposes: it verifies that our permittivity measurements are robust across analytical frameworks and instrumental platforms; it demonstrates that the canonical Gaussian fit, which encodes only the peak emission wavelength, can be matched by a fit-free, two-dimensional spectral phasor approach capturing both spectral redshift (phase) and broadening (modulus, reflecting quenching and spectral quality); and it shows that the phasor method can be applied to more complex spectra where simple Gaussian fitting is insufficient,

enabling straightforward permittivity mapping in challenging systems[14].

Additionally, Fig. 1e,f demonstrates that the redshift of ACDAN's emission spectrum is governed by a single physical quantity, namely the solvent dielectric permittivity. Importantly, this empirical observation is fully consistent with first-principles fluorescence theory, where according to the Lippert–Mataga relation[37], dipole–dipole interactions between the fluorophore and its dielectric environment give rise to a predictable shift in the emission maximum (see SI). Indeed, the Lippert–Mataga equation accurately reproduces the calibration data obtained across a broad range of solvents (Fig. 1e, f solid curve), establishing a robust relationship between the maximum

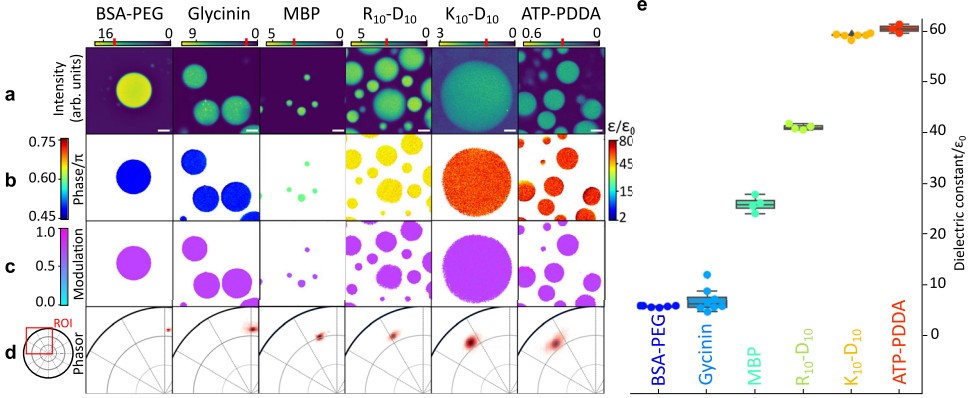

**Fig. 2 | ACDAN hyperspectral imaging enables the broad-range determination of local permittivity in the dense phase of biomolecular condensates.**
**a** Representative mean intensity images (averaged over the entire detected spectral range) of various biomolecular condensate systems, as labeled. Upper color bars indicate fluorescence intensity (arbitrary units), with red marks denoting the pixel exclusion threshold applied to (**b**–**d**). Scale bars: 5 μm. The number of repetitions is given in (**e**). **b** Local dielectric permittivity maps (right color bar) derived from the spectral phase (left color bar). The spectral phase, which roughly reflects the emission maximum (see Fig. 1d and "Methods"), is translated into permittivity values using the microscopy-based phasor calibration curve (Fig. 1f). **c** Maps of spectral modulus (left color bar), reflecting spectrum broadness (Fig. 1c, "Methods"), showing only minor differences across condensate systems. **d** Phasor plot

representation of the spectral information. Each dot represents the phase and modulus of an individual pixel from the image stacks in (**a**). The left inset highlights the region of interest (ROI) covering the experimental permittivity range (from $2\varepsilon_0$ to $80\varepsilon_0$). The spread of the pixel cloud illustrates the measurement error.
**e** Dielectric permittivity values for all imaged condensate systems span a surprisingly broad range from ~$5\varepsilon_0$ (comparable to oil-like environments) to ~$60\varepsilon_0$ (closer to water) as sensed in the immediate environment of ACDAN. Each dot represents data from a stack of images. Boxplots show median (central line) and interquartile range (box); whiskers extend to the furthest datapoint within 150% of the interquartile range (BSA-PEG and $K_{10}$-$D_{10}$: $n = 6$, glycinin: $n = 8$, ATP-PDDA: $n = 3$, $R_{10}$-$D_{10}$ and MBP: $n = 4$); see Fig. S4 for additional statistical details and experimental conditions.

emission wavelength $\lambda_{max}$ and solvent permittivity $\varepsilon$:

$$\frac{1}{\lambda_{max}} \sim \mu_E \frac{(\mu_G - \mu_E)}{V_{dye}hc} f(\varepsilon) + \text{const}, \qquad (1)$$

where $\mu_E$ and $\mu_G$ are the excited state and ground state dipole moments of the fluorophore, $V_{dye}$ is the effective molecular volume of ACDAN, $h$ is the Planck constant, $c$ is the speed of light in vacuum, and $f(\varepsilon)$ is the generalized Debye function (see SI for full model description). While Eq. (1) accurately describes the hyperspectral imaging data (Fig. 1e, f), it shows discrepancies with the spectrofluorimetry data, for which an empirical logarithmic function provided a better fit (see Fig. S1d and SI text for discussion).

Taken together, our results demonstrate that ACDAN emission provides a sensitive and broadband readout of dielectric permittivity ($2 - 80\varepsilon_0$). The close agreement between our calibration and validation datasets (Fig. 1e, f) demonstrates high sensitivity that is consistent across different solvent chemistries and independent of the measurement platform (microscopy or spectrofluorimetry). In these systems, ACDAN behaves as a passive solvatochromic probe, allowing us to treat the dye emission redshift, quantified by $\lambda_{max}$, as a bijective function of the permittivity of the dye immediate microenvironment: each $\lambda_{max}$ corresponds to a unique permittivity $\varepsilon$, and vice versa. While we expect this relationship to hold broadly, potential limitations in systems containing folded proteins with specific binding sites are addressed in the "Discussion" section.

In the following, we employ the microscopy-based hyperspectral imaging approach for data acquisition and the spectral phasor-based framework (Fig. 1c) to resolve biomolecular condensates permittivity (calibration detailed in Fig. S2). We chose this framework for two key reasons: first, its lack of requirement for an analytical fit function, especially relevant for probes with complex emission spectra (e.g., LAURDAN[14] or multifunctional fluorescent probes[21,39]) that are not accurately described by Gaussian models; second, it exhibits superior accuracy, with a maximum error of 8% compared to 13% for Gaussian fitting (Table S3).

## Hyperspectral imaging allows determination of biomolecular condensates permittivity

To evaluate condensate permittivity, we analyzed six different systems, for which three distinctive types of LLPS can be identified: (i) segregative LLPS in polyethylene-glycol (PEG)–bovine serum albumin (BSA), and PEG–dextran solutions, (ii) self-coacervation in myelin basic protein (MBP) and soy-plant based glycinin condensates, and (iii) complex coacervation in adenosine triphosphate (ATP)–polydiallyldimethylammonium chloride (PDDA), and oligopeptides-based condensates made of poly-lysine (K)–poly-aspartic acid (D) ($K_{10}$-$D_{10}$) and poly-arginine (R)–poly-aspartic acid (D) ($R_{10}$-$D_{10}$). Among these, the PEG-dextran, glycinin, and PEG-BSA systems were chosen for their cost-effectiveness and the feasibility of physically separating and analyzing their individual coexisting phases, facilitating more systematic characterization. The results of the phasor-based analysis of hyperspectral stacks for these systems are presented in Fig. 2. In this figure, the 2D phasor representation allows us to precisely distinguish between two metrics of interest encoded in the spatial coordinates; the *phase* reports on ACDAN redshift and encodes condensate permittivity (panel b) and the *modulus* reports on spectral sharpness and encodes dye emission quenching (panel c).

Panels (**b**–**d**) of Fig. 2 were produced by applying a minimal intensity threshold to identify the highest intensity portion of the data in panel (**a**). This procedure distinguished condensates already settled at the bottom of the imaging chamber (highest mean signal) from those sedimenting into the observation plane during acquisition (moderate to low mean signal). The latter exhibited artificially elevated permittivity due to the absence of shorter-wavelength spectral components and were therefore excluded from the analysis as indicated in Fig. 2a.

We also observed that BSA-PEG condensates display higher fluorescence intensity, likely reflecting an increased quantum yield of ACDAN and a known tendency of the "sticky" BSA protein to bind small molecules[40]. As a solvatochromic fluorophore, ACDAN probes the dielectric properties in its immediate molecular vicinity rather than a bulk-averaged environment; in this case, the enhanced signal suggests

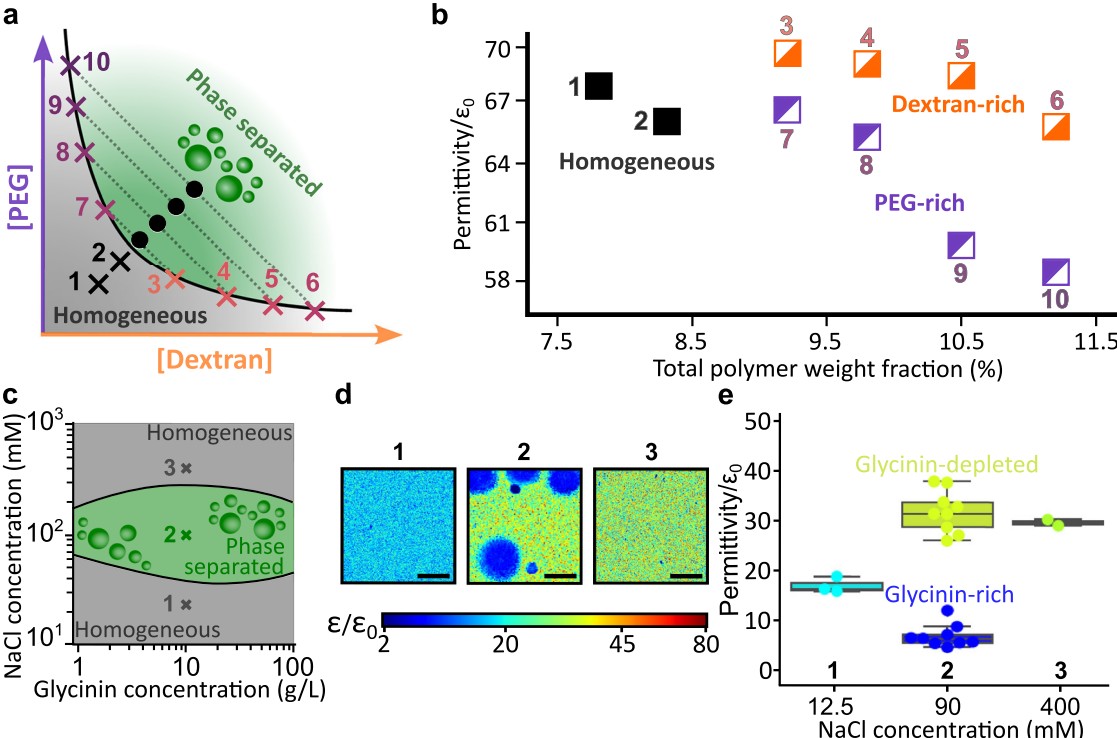

**Fig. 3 | Dielectric permittivity mapping of coexisting phases across PEG-dextran and glycinin phase diagrams. a** Sketch of the PEG-dextran phase diagram, see ref. 44 for experimental values. Crosses (1–2) represent homogeneous PEG-dextran solutions. Black dots indicate the total polymer concentration of the tested phase-separated mixtures. The binodal is shown by a black curve and grey dotted lines represent the tie lines. The endpoints of the tie lines indicate the dextran-rich phase compositions (crosses 3–6) and the PEG-rich phase compositions (7–10). **b** Permittivities of homogeneous and phase separated solutions at different total polymer concentration. The labels next to the data points correspond to the compositions schematically presented in (**a**), see Fig. S5 for hyperspectral maps and Table S4 for exact compositions. **c** Phase diagram of glycinin condensation as a function of protein and NaCl concentrations[68]. Grey regions indicate homogeneous mixtures and the green region denotes phase separation (condensate formation). Points 1, 2, and 3 correspond to NaCl concentrations of 12.5, 100, and 400 mM, respectively, at a constant protein concentration of 10 g/L. **d** Example permittivity maps for glycinin solutions at three NaCl concentrations indicated in (**c**). Color bar: rescaled permittivity. Scale bars: 10 μm. **e** Quantified permittivity of all phases at points 1, 2, and 3 in (**c**, **d**). Boxplots show median (central line) and interquartile range (box); whiskers extend to the furthest datapoint within 150% of the interquartile range (point 1: $n = 3$, point 2: $n = 9$, and point 3: $n = 2$).

that the measured permittivity predominantly reflects protein-proximal microenvironments rather than a simple bulk average (see "Discussion").

Figure 2b, c shows that while phase values (encoding dye redshift, Fig. S2), and thus permittivity (Fig. 2b), vary significantly among condensate systems, the modulus (related to spectra sharpness, Fig. S2) only slightly decreases with increasing permittivity (Fig. 2c). This observation is further evident in Fig. 2d and consistent with the expected fluorescence quenching effect associated with dipolar relaxation[37]. The spread of the pixel cloud in Fig. 2d visually illustrates measurement error, which is inversely proportional to the signal-to-noise ratio.

Averaging the phase values and converting them to permittivity yields Fig. 2e (additional statistical details in Fig. S4). The data reveal a remarkably broad range of dielectric constants across different condensates, from values as low as ~5$\varepsilon_0$ (typical of oil-like environments) to as high as ~60$\varepsilon_0$ (approaching that of water), highlighting substantial variability in condensate hydrophobicity as sensed by ACDAN. Additionally, the weak correlation between the type of interaction driving LLPS and condensate permittivity (Fig. 2e) suggests that condensate dielectric properties cannot be predicted solely based on the type of LLPS mechanism. A striking example is the comparison between $K_{10}$-$D_{10}$ and $R_{10}$-$D_{10}$ condensates both formed via similar electrostatic heterotypic interactions; despite only four-atom difference between lysin and arginine, their permittivities differ drastically, i.e., by ~20$\varepsilon_0$ (Fig. 2e). These observations are consistent with in silico

assessments of differential hydration energies in arginine- versus lysin-rich assemblies[41,42].

## Dielectric permittivity as a descriptor of dense and dilute phases

Changes in macromolecules concentration within either the dense or depleted phase modulate protein hydration and water activity, thereby altering the permittivity contrast between the two phases[2,11,43]. Thermodynamically, such shifts reflect changes in the system position on its phase diagram[44]. From this perspective, measuring the permittivity contrast between two coexisting phases provides a qualitative readout of a mixture's phase state. Hyperspectral imaging of ACDAN as applied here, enables simultaneous determination of the permittivity of coexisting phases, from which changes in the position of a mixture within its phase diagram can be inferred, as illustrated in Fig. 3.

As a benchmark system, we first examined a canonical aqueous two-phase system (ATPS) composed of PEG and dextran. At low polymer concentrations, they form a homogeneous aqueous solution, while increasing the concentration of either polymer induces LLPS into a PEG-rich phase and a dextran-rich dense phase (Fig. 3a). This PEG-dextran ATPS is a well-established model system with robustly characterized phase diagrams (see e.g., ref. 45), making it ideally suited to validate our approach and to systematically track how dielectric permittivity varies with macromolecular composition.

Although water is present in both phases (making this formally a ternary system), the PEG-dextran phase diagram and tie lines provide a way to evaluate the water content and macromolecular crowding in

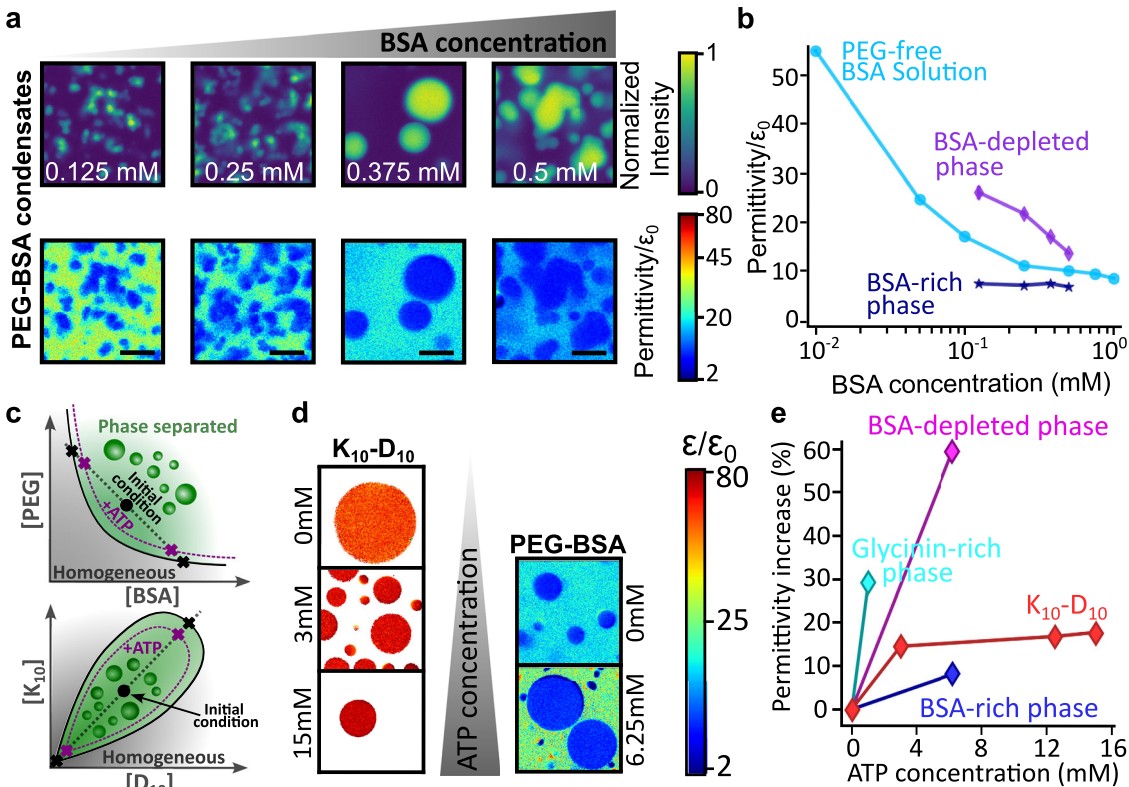

**Fig. 4 | Evolution of dielectric permittivity with macromolecule concentration and ATP modulation. a** Effect of BSA concentration on the permittivity of PEG-BSA condensate systems. Top: average intensity projections of hyperspectral stacks for different initial BSA concentrations (indicated in white). Bottom: corresponding permittivity maps of the same regions, showing a gradual decrease in the depleted phase permittivity with increasing protein concentration. Scale bars: 10 μm.
**b** Increasing the total BSA concentration lowers the permittivity of homogeneous protein solutions in water (blue circles) and BSA-depleted phase (diamonds), but has no significant effect on the PEG-BSA condensates (stars); see Fig. S6 for the respective permittivity maps. **c** Schematic phase diagrams of the PEG–BSA system (top) and the $K_{10}$-$D_{10}$ system (bottom). The solid black curves denote binodals; black dotted lines represent tie lines through given initial compositions (black dots). Black crosses indicate the compositions of the coexisting dense and depleted

phases for these initial compositions in the absence of ATP. The purple dotted curves illustrate the expected shift of the binodal upon ATP addition, reflecting weakened intermolecular interactions. Correspondingly, purple crosses indicate the altered compositions of the coexisting phases in the presence of ATP.
**d** Permittivity maps illustrating the effect of increasing ATP concentration on $K_{10}$-$D_{10}$ and BSA-PEG condensates. Each frame represents a $36.9 \times 36.9$ μm² sample area. The color bar (logarithmic scale) indicates permittivity. Left: permittivity maps of $K_{10}$-$D_{10}$ condensates showing increasing permittivity with ATP concentration. Right: permittivity maps of BSA-PEG condensates (10 wt% PEG, 0.375 mM BSA) demonstrating a similar trend. **e** Impact of ATP concentration on the permittivity of $K_{10}$-$D_{10}$, and BSA-PEG condensates (shown in **d**) and glycinin condensates (10 g/L total protein in 100 mM NaCl). Permittivity values in the absence of ATP are given in Fig. 2.

each phase based on changes in polymer concentration. Building on this knowledge, Fig. 3b illustrates how variations in the initial PEG and dextran concentrations, schematically indicated as points in the phase diagram (Fig. 3a), translate into distinct permittivity values.

Measurements for points (3–10) in Fig. 3a, b were performed by isolating the PEG-rich and dextran-rich phases of the ATPS; corresponding permittivity maps and phase compositions are provided in Fig. S5 and Table S4. Even a small increase in polymer concentration (a few tens of percent), accompanied by a reduction in water fraction, consistently results in lower permittivity, corroborating previous reports[46]. Moreover, Fig. 3a, b shows that changes in total polymer concentration (crosses 1 and 2 for homogeneous and black dots for phase-separated solutions in panel a) result in different levels of macromolecular crowding and water content in both homogeneous and phase-separated regimes, which are quantitatively reflected in the measured permittivity values.

Figure 3c–e shows that different points in the glycinin phase diagram (Fig. 3c) correspond to markedly different permittivity values between homogeneous and phase-separated states. Figure 4 further emphasizes the generality of these trends by showing similar results for PEG-BSA systems. Figure 4a shows that increasing the overall BSA concentration decreases the permittivity of the BSA-depleted phase, a

trend quantified in Fig. 4b and compared to PEG-free BSA aqueous solutions.

Overall, Figs. 3 and 4a, b establish that changes in macromolecular crowding systematically translate into variations in permittivity sensed by ACDAN across a range of phase-separated systems, with magnitudes that depend strongly on the molecular nature of the macromolecules. In segregative phase separation, increasing PEG or dextran concentrations (Fig. 3a, b), as well as increasing BSA concentration (Fig. 4a, b), enhance macromolecular crowding, leading to higher polymer or protein concentrations in both coexisting phases. These changes are consistently reflected by a reduction in the measured permittivity in each phase.

Similarly, Fig. 3c–e shows that increasing NaCl concentration above the lower critical point of glycinin solutions induces protein condensation, leading to a pronounced increase in macromolecular crowding within the dense phase, which is accompanied by a marked decrease in permittivity. Importantly, Fig. S7 presents spectrofluorimetric controls demonstrating that NaCl concentration alone does not substantially affect ACDAN fluorescence. Consistent microscopy-based controls confirming the salt insensitivity of ACDAN emission have been reported previously[14]. Together, these results demonstrate that macromolecular crowding and ionic conditions

influence ACDAN emission indirectly, through their impact on the local dielectric environment sensed by the probe.

Permittivity measurements in PEG–BSA phase-separated systems further illustrate how the dielectric microenvironment differs between condensed and depleted phases. In these mixtures, both protein and polymer concentrations vary simultaneously across coexisting phases, and PEG itself strongly influences the permittivity (Fig. 3a, b). Thus, permittivity values as sensed by ACDAN reflect the combined effects of water content, macromolecular crowding, but, importantly, also local probe–microenvironment interactions as we discuss below.

To further assess the sensitivity of our method to subtle physicochemical changes, we investigated the effects of ATP as a hydrotrope (a small molecule known to enhance protein solubility and hydration[47–49]) and examined its effect on the dielectric permittivity of various condensate systems. Figure 4c schematically summarizes the expected effects of ATP addition on the phase behavior of the investigated systems. ATP acts to weaken the intermolecular interactions that drive phase separation, resulting in a lower macromolecule concentration (and correspondingly higher water content) within the dense phase, alongside an increased protein/polymer concentration in the depleted phase. In the phase diagram representation, this manifests as a shift of the binodal boundaries away from the origin upon ATP addition, consistent with the observed reduction in permittivity contrast between coexisting phases as shown in Fig. 4d, e.

These results underscore the sensitivity of this method in detecting subtle variations in hydration and microenvironment within both condensed and depleted phases. Specifically, adding just a small amount of ATP progressively increases the permittivity of glycinin and $K_{10}$-$D_{10}$ condensates by approximately 25% and 15%, respectively (Fig. 4d). Similar trends are observed in the BSA-PEG system, where the depleted phase permittivity increases by 60% upon addition of 6.25 mM ATP, compared to a modest 8% increase for the condensed phase. These differential responses suggest that ATP shifts the system's position on its phase diagram (Fig. 4c) by selectively altering the hydration and dielectric permittivity of the coexisting phases.

Altogether, these findings demonstrate that permittivity measurements provide a sensitive readout of changes in dielectric microenvironment within coexisting phases, enabling qualitative tracking of how phase behavior responds to molecular perturbations such as ATP, without relying on assumptions about phase composition.

## Permittivity contrast between dense and depleted phases governs condensate-membrane interactions

Building on the hypothesis that condensate hydrophobicity governs interactions with cellular structures like membranes, we investigated whether dielectric properties directly influence condensate-membrane affinity. Previous work shows that condensate-membrane interactions vary with bilayer hydrophobicity, modulated by, for example, lipid tails saturation, even when membrane surface chemistry (lipid headgroups) remains unchanged[35]. This intuitively suggests that condensate hydrophobicity, quantified through permittivity, may govern this interaction.

Upon contacting a membrane, condensates undergo a wetting transition reflecting the strength of their interaction[50]. This leads to mutual reshaping of both the membrane and the condensate[51], where the resulting equilibrium morphology directly reflects the interaction affinity. Figure 5a, b presents examples of such morphologies, observed during the interaction of condensates with membranes[52], exemplified here by giant unilamellar vesicles, and well-described by the theoretical framework introduced in ref. 51.

The force balance between the tensions of the wetted ($\Sigma_{ic}^m$) and bare ($\Sigma_{ie}^m$) membrane segments and the condensate interfacial tension ($\Sigma_{ce}$)[51,53] (see "Methods") shapes the vesicle-condensate couple into an axisymmetric system characterized by three apparent contact angles ($\theta_i, \theta_c, \theta_e$) (Fig. 5a and "Methods"). These apparent contact angles are

extracted from confocal cross-sections passing through the symmetry axis of the vesicle–condensate system. While they fully describe the observed geometry, their values depend on extrinsic parameters such as condensate and vesicle size, as well as the degree of vesicle deflation, which vary between individual vesicle–condensate pairs in a sample. By contrast, at the nanometer scale, the membrane is smoothly curved at the three-phase contact line (black circle in Fig. 5a)[54] and wetting is characterized by the intrinsic contact angle $\theta^{in}$ [55]. This intrinsic angle is a material parameter that is independent of system geometry or size[51] and quantifies the membrane preferential affinity for the protein-rich (condensed) versus protein-poor (depleted) phase[51].

While $\theta^{in}$ has been shown to vary with salt, polymer or protein concentration[14,44,50], membrane charge[51] and lipid composition[35], the fundamental basis underlying these variations remains unclear. Intuitively, one might expect that a condensate with a permittivity similar to that of the membrane would exhibit stronger affinity. However, the intrinsic contact angle reflects not the absolute affinity of one phase for the membrane, but rather the relative affinities of the two coexisting aqueous phases, see "Methods". Because the sum of the contact angles opening toward each phase must equal 180° (see inset in Fig. 5a), a change in membrane preference for one phase necessarily implies a corresponding change for the other.

As a first step to test this hypothesis, we quantified the permittivities of coexisting phases in heterogeneous glycinin solutions and PEG-dextran ATPS. While the permittivity of the glycinin-depleted phase exhibited decrease markedly with increasing salt concentration, the condensed phase remained unaffected (Fig. 5c). This led to a monotonic increase in the permittivity contrast ($\varepsilon_c - \varepsilon_d$), see Fig. 5d (bottom). For PEG-dextran mixtures, increasing polymer concentration caused both phases to become less polar, with a stronger permittivity drop in the PEG-rich phase compared to the dextran-rich droplet phase (Fig. 3b), resulting in an increase in permittivity contrast ($\varepsilon_c - \varepsilon_d$) with increasing overall polymer concentration (see Fig. 4d, top panel). The trends in Fig. 5d show that permittivity contrast can be precisely tuned by modulating external variables such as ionic strength or polymer concentration.

We then examined whether this permittivity contrast correlates with condensate-membrane wetting. Figure 5e reveals a linear dependence of the intrinsic contact angle on ($\varepsilon_c - \varepsilon_d$) across independent $\theta^{in}$ datasets for various membrane types and condensate systems taken from refs. 44,51. Glycinin condensates interacting with DOPC:DOPS vesicles (orange data) exhibit an increasing $\theta^{in}$, and thus higher membrane affinity, as NaCl concentration rises, which enhances the permittivity contrast. In contrast, at 100 mM NaCl, increasing the charged lipid content (10, 20, and 40 mol% DOPS shown in rose, vermilion, and red) reduces condensate affinity for the membrane. For PEG-dextran mixtures, interaction with either liquid-disordered (DOPC:DPPC:Chol, 64:15:21, blue data in Fig. 5e) or liquid-ordered (DOPC:DPPC:Chol, 13:44:43, green) membranes show that increasing polymer concentration, which enhances the permittivity contrast, systematically increases membrane-condensate affinity as quantified by $\theta^{in}$. This universal linear correlation suggests a fundamental biophysical mechanism where the membrane preference between condensed and depleted phases depends primarily on their permittivity contrast.

In first approximation, this trend could be understood considering the membrane adhesion free energies to the two aqueous phases: $W_{mc}$ for the condensate and $W_{me}$ for the depleted phase (see "Methods"). Assuming each adhesion energy depends on the permittivity contrast between the membrane and the respective aqueous phase, ($\varepsilon_c - \varepsilon_m$) and ($\varepsilon_d - \varepsilon_m$), the affinity contrast $W = W_{mc} - W_{me}$ becomes proportional to $\varepsilon_c - \varepsilon_d$. Thus, the intrinsic contact angle, which encodes this affinity contrast, naturally reflects the dielectric difference between the two phases. Notably, while the

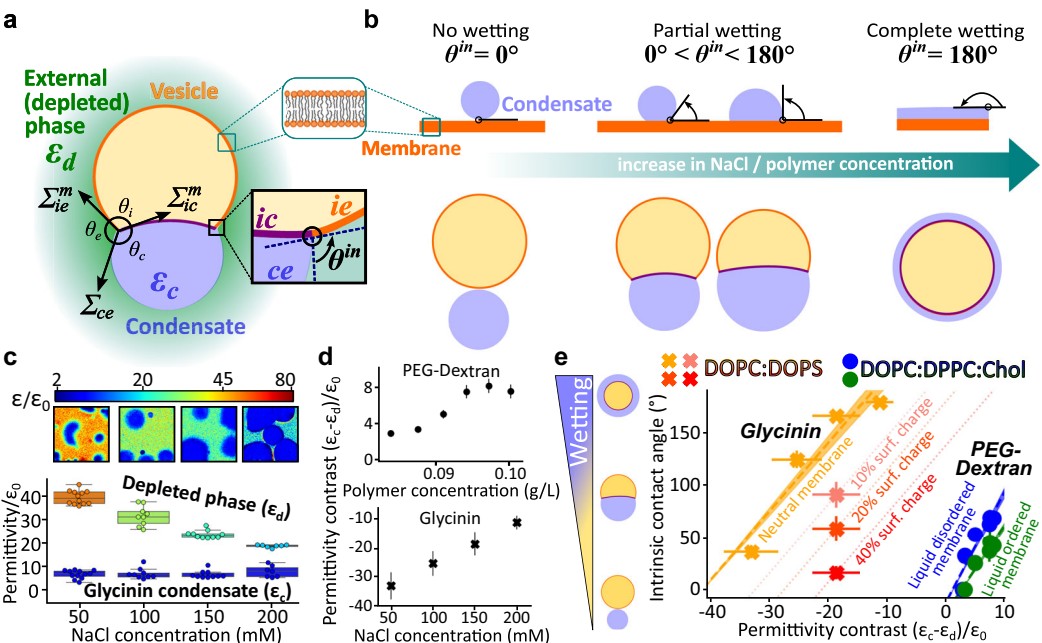

**Fig. 5 | Membrane wetting by condensates is modulated by the permittivity contrast between coexisting phases. a** Schematic of a giant unilamellar vesicle (yellow) wetted by a condensate (blue, permittivity $\varepsilon_c$) and surrounded by the polymer-depleted phase (light green, permittivity $\varepsilon_d$). The apparent contact angles facing the vesicle interior ($\theta_i$), the external phase ($\theta_e$), and the condensate ($\theta_c$), define the equilibrium morphology. Measured from confocal cross-sections, these angles reflect the balance of interfacial forces at the three-phase contact line (black circle), including the condensate interfacial tension $\Sigma_{ce}$ and the mechanical tensions of the two membrane segments contiguous to the depleted phase ($\Sigma_{ic}^m$, orange) and condensate phase ($\Sigma_{ic}^m$, purple). At the nanometer scale, the membrane does not exhibit a kink but is smoothly curved (inset). Wetting is described by the intrinsic contact angle $\theta^{in} = \arccos\left(\frac{\sin\theta_e - \sin\theta_c}{\sin\theta_i}\right)$ [51], see "Methods". **b** Schematics of condensate wetting flat lipid membranes (top) and vesicles (bottom) illustrating affinity regimes quantified by $\theta^{in}$ and characteristic morphologies. Increasing polymer or salt concentration enhances membrane affinity of dextran and glycinin

condensates[44,51]. **c** Hyperspectral permittivity maps of glycinin condensates at increasing NaCl concentrations. Top: dielectric maps of fields of view ($36.9 \times 36.9\ \mu m^2$). Bottom: boxplots showing that increasing salt primarily lowers the permittivity of the depleted phase, while the condensate phase is largely unaffected. Central lines: median; box: interquartile range (IQR); whiskers: $1.5 \times$ IQR (50 mM: $n = 12$, 100 mM: $n = 9$, 150 mM: $n = 10$ and 200 mM: $n = 7$). **d** Permittivity contrasts ($\varepsilon_c - \varepsilon_d$) between condensate and depleted phases as a function of polymer concentration (PEG-dextran, top, from Figs. 3b and S5), and ionic strength (glycinin-NaCl, bottom, from (**c**)). Error bars indicate mean $\pm$ SD. Datapoints for PEG-dextran (top) were extrapolated from ref. 44. **e** The intrinsic contact angle $\theta^{in}$, increases linearly with the permittivity contrast ($\varepsilon_c - \varepsilon_d$), indicating that stronger permittivity differences enhance membrane wetting as illustrated schematically on the left. Dashed lines are linear fits. Crosses: glycinin-membrane $\theta^{in}$ under different NaCl concentrations and membrane charge (ref. 51); circles: PEG-dextran ATPS at varying polymer concentration (ref. 44); see text for details.

intrinsic contact angle defining the membrane-condensate equilibrium topology explicitly accounts for the influence of condensate surface tension (see "Methods"), it is independent from condensate viscosity, which only influences the timescale over which the membrane-condensate system relaxes to its equilibrium configuration. As we see in Fig. 5e, the slope and offset of this linear relation are influenced by the membrane lipid composition, surface charge, and the specific condensate system.

Overall, these results underscore the crucial role of permittivity contrasts in modulating surface affinities between membranous compartments and protein-rich droplets. They highlight that condensate-membrane interactions cannot be understood by considering the dielectric properties of the condensates alone; rather, it is the permittivity difference between the condensed and depleted phases that governs affinity. Significantly, the ability of hydrotropes like ATP to differentially modulate the permittivity of coexisting phases (Fig. 4c, d) offers a mechanism to precisely tune condensate recruitment or repulsion from cellular membranes.

These findings call for systematic assessment of both phases when probing condensate interactions with biological materials. Hyperspectral permittivity mapping thus emerges as a powerful approach for understanding the physical principles that guide biomolecular condensation and its functional coupling to membrane systems.

## Discussion

The dielectric permittivity of biomolecular condensates is increasingly recognized as a key parameter linking their biophysical properties to physiological function[2,7,11]. Notably, permittivity differences between immiscible biopolymer phases govern their interfacial surface tension[2,11,17], which in turn stabilizes complex core-shell architectures in nuclear organelles[2]. This interdependence fundamentally arises from a water activity gradient that imposes an entropic penalty on interfacial water molecules[43,56,57]. Consequently, LLPS enables cells to locally regulate intracellular water activity and internal permittivity, acting as a feedback mechanism to buffer against osmotic or thermal stress[7,58]. Complementary preprint work has explored the internal dynamics of natural protein condensates[59], highlighting the broader relevance of condensate microenvironments.

Our study introduces a powerful hyperspectral imaging approach using the environment-sensitive fluorophore ACDAN to quantitatively map local dielectric permittivity within biomolecular condensates and their surrounding environment. This parameter, while crucial for understanding phase behavior, has remained experimentally elusive, especially in complex, heterogeneous systems where high resolution and dynamic range are essential. Our method provides robust, broadband measurements (2–80$\varepsilon_0$) independent of solvent chemistry or instrumental platform. Notably, we reveal a very wide range of

dielectric permittivities across different condensates, spanning from oil-like ( ~$5\varepsilon_0$) to water-like ( ~$60\varepsilon_0$), see Fig. 2e, highlighting substantial variability in their hydrophobicity. Importantly, we find that condensate permittivity is not solely dictated by the type of underlying LLPS interactions (Figs. 2–4), but is also strongly influenced by the protein-to-water ratio, i.e., water content, within the condensed phase (Fig. 3b, S5). To confirm this, we performed Raman spectroscopy measurements (Fig. S9), which revealed that higher-permittivity condensates like $K_{10}$-$D_{10}$ contain more water relative to peptide compared to condensates characterized by lower permittivity like $R_{10}$-$D_{10}$ (Fig. 2); we explore this further in a follow up study[60].

As a note of caution, while water content is a primary determinant of condensate permittivity, additional effects can dominate in systems containing folded proteins with hydrophobic microcavities, as exemplified by BSA. In homogeneous BSA solutions, we measure apparent permittivity values as low as $\varepsilon \approx 25\varepsilon_0$ at concentrations of ~0.05 mM. While reduced water dipolar relaxation in protein hydration layers can partially account for this effect[61], it cannot explain the nearly order-of-magnitude decrease observed at such low protein volume fractions.

We therefore propose that ACDAN partially binds to hydrophobic pockets within folded BSA, consistent with BSA's known affinity for small ligands. In such environments, water is highly structured and exhibits an effective permittivity of $\varepsilon \approx 5$–$6\varepsilon_0$[40]. Dye molecules bound to these pockets are expected to display enhanced quantum yield[40] (consistent with the markedly higher ACDAN fluorescence intensity in Fig. 2a), causing protein-associated ACDAN to dominate the fluorescence signal and, respectively, the permittivity readout.

In this regime, ACDAN reports the dielectric properties of its local protein-proximal microenvironment rather than the bulk-averaged permittivity reflecting the water content. Nevertheless, the dielectric differences observed in Figs. 4 and S8 are biologically relevant, as enzymatic activity requires a highly hydrated environment capable of sustaining a hydrogen-bonded water network at the protein surface[62,63]. In this context, the reduced permittivity measured in the condensed phase of PEG-BSA systems aligns with reports of decreased enzymatic activity in the condensed phase of PEG-BSA mixtures compared to homogeneous BSA solutions of similar overall concentrations[64].

We note that in condensates containing folded proteins, ACDAN signals may reflect local interactions with hydrophobic pockets rather than bulk solvent permittivity, and future applications of this method should consider that low measured permittivity may arise from structured protein-proximal environments rather than reduced water content. However, we emphasize that local, protein-proximal sensing of dielectric permittivity by ACDAN is specific to folded proteins with well-defined hydrophobic cavities, and is not observed for unstructured polypeptides ($K_{10}$, $R_{10}$, $D_{10}$) or hydrophilic polymers (PDDA, PEG, dextran), where condensate permittivity is governed primarily by the water-to-polymer ratio (Figs. 3b and S5) and is independent of dye concentration (for details see ref. 60).

Importantly, this observation highlights that dielectric permittivity in condensates and solutions of highly structured proteins can be very heterogeneous at the molecular scale and that local protein-proximal environments may differ substantially from expectations based on average composition alone. ACDAN-based measurements, therefore, provide access to the dielectric microenvironments experienced by proteins and small molecules (i.e., environments that are likely most relevant for biochemical function), even when these differ from macroscopic expectations and bulk hydration.

Beyond hydration and phase behavior, our high-resolution approach, which allows simultaneous measurement of both condensed and depleted phases, provides important complementary insights into the interactions of condensates with membranes. We find that it is not the absolute permittivity of condensates, but rather the permittivity contrast with their environment, that is the central parameter controlling interactions with lipid membranes. A key result of our study is the discovery of a universal linear relationship between the intrinsic contact angle, which characterizes membrane-condensate affinity, and the permittivity contrast between the coexisting phases (Fig. 5e). While many studies focus solely on the properties of the condensate phase, our findings underscore the critical importance of also characterizing the protein-poor condensate environment.

It is important to distinguish changes in condensate-membrane affinity from variations in the free energy of demixing. If these quantities were directly coupled, one would expect the intrinsic contact angle in the glycinin-membrane system to exhibit a maximum at the point of largest separation from both the upper and lower binodals in Fig. 3c, where the thermodynamic driving force for phase separation is maximal. Instead, the intrinsic contact angle increases monotonically with NaCl concentration[51]. This behavior is consistent with the observed evolution of the permittivity contrast: while the permittivity of the dense phase remains largely insensitive to salt, that of the dilute phase decreases continuously (Fig. 5c). These trends indicate that membrane affinity is governed by phase-specific dielectric properties rather than by the demixing free energy alone. Additional contributions may arise from NaCl-dependent changes in protein conformation, as suggested previously[14].

Beyond thermodynamic considerations, membrane-condensate interactions emerge from a multifaceted interplay of molecular polarity, surface orientation, charge, and ion partitioning. While previous studies have explored the effect of condensate surface potential, ionic strength[51] and protein sequence[36], our results extend this framework by identifying dielectric permittivity as a quantitative, bulk parameter that directly governs condensate-membrane interactions, revealing an additional and previously unrecognized physical determinant of wetting behavior.

Given that the hydrophobic core of lipid membranes exhibits a low permittivity (comparable to oils), condensates with inherently low dielectric permittivity are likely to exhibit greater affinity to membranes. At comparable permittivity contrasts, we speculate that such condensates, like glycinin and BSA-PEG (Fig. 2e), will display enhanced membrane wetting compared to more hydrophilic, high-permittivity condensates, such as $K_{10}$-$D_{10}$, ATP-PDDA, and the dextran-rich phase in PEG-dextran ATPS (Figs. 2e and 3b). The membrane surface charge further modulates this interaction: achieving the same intrinsic contact angle on charged membranes requires a higher permittivity contrast than on neutral membranes (Fig. 5e). Together, these observations indicate that membrane-condensate interactions arise from competing contributions: for highly charged membranes, surface electrostatics dominate and largely mask dielectric effects, whereas for weakly charged membranes, the bulk permittivity contrast between dense and dilute phases becomes the primary determinant of condensate affinity. Similarly, more ordered (and presumably more hydrophobic due to higher lipid packing) membranes demand larger permittivity contrasts to reach comparable wetting, likely due to their lower intrinsic permittivity.

We also propose that lower-permittivity condensates may preferentially stabilize hydrophobic membrane defects like pores, a property with potential relevance for cellular membrane repair. In this context, condensates can act as plugs for transient membrane pores, helping to stabilize and reseal damaged membrane organelles[5,65]. Condensates with lower permittivity may likely allow better integration with the hydrophobic membrane environment, facilitating pore sealing and restoring membrane integrity.

Beyond these fundamental insights, our method provides a sensitive dielectric descriptor of coexisting phases across different regions of a phase diagram (Figs. 3 and 4). While it does not yield a direct readout of phase composition, it robustly captures relative changes in the physicochemical environments of dense and dilute phases, allowing the quantification of subtle changes induced by

hydrotropes like ATP (Fig. 4c, d). Considering ATP's central role in cellular physiology, our results suggest that ATP not only influences protein solubility[34,48] but also acts as a key modulator of condensate interactions with other cellular components, by affecting permittivity contrast between coexisting phases.

Overall, our results underscore the utility of this method in mapping fine variations in phase behavior, composition, and dielectric properties with high spatial resolution and across diverse biomolecular condensate systems. We believe these findings significantly advance the physical framework for understanding the interplay between electrostatics, phase separation, and biological function in complex and heterogeneous biomolecular systems and in understanding condensate interactions. Given the accessibility of hyperspectral imaging on commercial confocal microscopes[66], the method is poised to become a valuable tool for studying dynamic intracellular environments and advancing the understanding of biomolecular phase transitions in health and disease. The approach paves the way for new investigations into the physicochemical mechanisms underlying condensate function, aging, and liquid-to-solid transitions associated with neurodegenerative pathologies[27,67].

# Methods

## Materials
6-acetyl-2-dimethylaminonaphthalene (ACDAN) was purchased from Santa Cruz Biotechnology (USA). PDDA (200–350 kDa, 20 wt% solution in water), ATP, and sodium hydroxide (NaOH) were obtained from Sigma-Aldrich (Missouri, USA). Poly(ethylene glycol) (PEG 8000, Mw 8 kg/mol and PEG 400, Mw 4 kg/mol) and dextran from Leuconostoc mesenteroides (molecular weight between 400 and 500 kDa) were purchased from Sigma-Aldrich. The oligopeptides, poly-L-lysine hydrochloride (degree of polymerization, $n = 10$; $K_{10}$), poly and poly-L-aspartic acid sodium salt ($n = 10$; $D_{10}$) were purchased from Alamanda Polymers (AL, USA) and used without further purification ($\geq 95\%$). BSA ($\geq 98\%$ purity, 66 kg/mol), MBP bovine ($\geq 90\%$), 1,4-Dioxane, Anisol (99%), 1-Hexanol, and 1-Butanol (99.9%) were purchased from Sigma-Aldrich. Mineral oil was purchased from Carl Roth GmbH. Ethanol absolute was purchased from VWR chemicals BDH. Phosphate-buffered saline (PBS) pH 7.2 was purchased from Thermofisher. Polyvinyl alcohol (Mw 145000 kg/mol) was purchased from Merck (Darmstadt, Germany). All solutions were prepared using ultrapure water from SG water purification system (Ultrapure Integra UV plus, SG Wasseraufbereitung) with a resistivity of 18.2 MΩ cm.

## Condensate formation and labelling
Coverslips for confocal microscopy (26 × 56 mm, Waldemar Knittel Glasbearbeitungs GmbH, Germany) were washed with ethanol and water, then passivated with a 10 mg/mL BSA solution and an aliquot of 10 μL of condensates suspension was placed on the coverslip before imaging.

**PDDA-ATP condensates.** Phase-separated droplets were formed by gently mixing aliquots of stock solutions of ATP and PDDA (in this order) with ACDAN in pure water to a final volume of 10 μL. The final concentration of each component was as follows: 14.8 mM ATP, 4.9 mM PDDA, 10 μM ACDAN.

**Glycinin condensates.** Freeze-dried glycinin was a gift from Dr. Nan-nan Chen. The purification is detailed in ref. 68. A 20 mg/mL glycinin solution at pH 7 was freshly prepared in ultrapure water and filtered with 0.45 μm filters to remove any insoluble materials. To form the condensates, the desired volume of the glycinin solution was mixed with the same volume of a NaCl solution of twice the desired final concentration. In this manner, the final protein concentration was 10 mg/mL[68]. The final concentration of ACDAN was 5 μM.

**$K_{10}$-$D_{10}$ and $R_{10}$-$D_{10}$ condensates.** Phase-separated droplets were formed by gently mixing aliquots of stock solutions of $D_{10}$ and $K_{10}$ or $R_{10}$ (in this order) with ACDAN in pure water to a final volume of 10 μL. The final concentration of each component was as follows: 2.5 mM $D_{10}$, 2.5 mM $K_{10}$ or $R_{10}$, 15 μM ACDAN.

**PEG-BSA condensates.** Phase-separated droplets were formed by gently mixing aliquots (via pipetting and releasing 3 times the total volume) in a 1:1 ratio of stock solutions of 20% PEG-8000 in PBS with 30 μM ACDAN with BSA dissolved in PBS at half the desired concentration and to a final volume of 10 μL.

**PEG-Dextran condensates.** A polymer solution in composed of the desired weight fractions of PEG and dextran were prepared and left, when relevant, for 2 days to completely phase separate and equilibrate. ACDAN was then added to each phase to reach a final concentration of 25 μM of ACDAN.

**MBP condensates.** MBP condensates were prepared by following the procedure described in ref. 69. Briefly, a 5 mg/mL solution of MBP dissolved in water was mixed with a 20 mM NaOH solution of 10 μM ACDAN in a 1:1 ratio.

## Confocal microscopy and hyperspectral imaging
Hyperspectral images were acquired using a confocal Leica SP8 FAL-CON microscope equipped with a 63×, 1.2 NA water immersion objective (Leica, Mannheim, Germany). The microscope was coupled to a pulsed Ti:Sapphire laser MaiTai (SpectraPhysics, USA), with a repetition rate of 80 MHz. A two-photon wavelength of 780 nm was used for ACDAN excitation. Image acquisition was performed with a frame size of 512 × 512 pixels[14,51] and a pixel size of 72 × 72 nm using a Hyd SMD detector in standard mode. For hyperspectral imaging, the $xy\lambda$ configuration of Leica SP8 was used, sequentially measuring in 32 channels with a bandwidth of 9.75 nm in the range from 416 to 728 nm. Some hyperspectral images were realigned using the ImageJ software and all hyperspectral stacks were processed by the SimFCS software developed at the Laboratory of Fluorescence Dynamics (available at https://www.lfd.uci.edu/globals/), and analyzed using Python code based in the PhasorPy library (available at https://www.phasorpy.org/).

## Gaussian fit analysis
The Gaussian fit analysis of ACDAN hyperspectral imaging data consists in approximating the wavelength-intensity profile of ACDAN emission by a skewed Gaussian function of the form:

$$I(\lambda) = I_0 \Psi\left(\frac{\lambda - \mu}{\sigma}\right) \Phi\left(\gamma\left(\frac{\lambda - \mu}{\sigma}\right)\right), \qquad (2)$$

$$\Psi(x) = \frac{1}{\sqrt{2\pi}} \exp\left(-\frac{x^2}{2}\right), \qquad (3)$$

$$\Phi(x) = \frac{1}{2}\left(1 + \mathrm{erf}\left(\frac{x}{\sqrt{2}}\right)\right) \qquad (4)$$

where $\Psi$, $\Phi$, erf, $\lambda$, $\mu$, $\sigma$, $I_0$ and $\gamma$ respectively represent the normal distribution function, the cumulative distribution function, the error function, the wavelength, the mean wavelength, the mean deviation, the intensity scaling factor and the skewness factor. Data interpolation was performed based on computer programing script written in Python and based on the scipy library. The $\lambda_{max}$ was then evaluated based on the position of the maximum of the fitted ACDAN emission curve and used as an input in the $\lambda - \varepsilon$ calibration curves presented in Figs. 1e and S1 in the main text.

## Spectral phasor analysis

The spectral phasor analysis of ACDAN hyperspectral imaging data consists in calculating the real and imaginary components of the Fourier transform, respectively referred to as G and S, and using them as Cartesian coordinates on a 2D spectral phasor map. (G, S) are defined by the following expressions:

$$G = \frac{\int_{\lambda_0}^{\lambda_f} I(\lambda) \cos(\omega n (\lambda - \lambda_0)) \, d\lambda}{\int_{\lambda_0}^{\lambda_f} I(\lambda) \, d\lambda} \tag{5}$$

$$S = \frac{\int_{\lambda_0}^{\lambda_f} I(\lambda) \sin(\omega n (\lambda - \lambda_0)) \, d\lambda}{\int_{\lambda_0}^{\lambda_f} I(\lambda) \, d\lambda} \tag{6}$$

where $I(\lambda)$ for a particular pixel represents the intensity as a function of wavelength, measured in the interval $(\lambda_0 ; \lambda_f)$. This range depends on instrumental constraints and the type of detector used for the analysis, in our case 416–728 nm. Note that changing the detection range will necessarily results in a change of the relative positions of different points on the phasor plot; therefore, the detection range must be conserved across all experiments in order to be able to compare measurements. The parameter $n$ is the harmonic, i.e., the number of cycles of the trigonometric function that are fit in the wavelength range by means of the angular frequency $\omega$:

$$\omega = \frac{2\pi}{(\lambda - \lambda_0)} \tag{7}$$

When imaging with a microscope, we acquire a discrete number of spectral steps corresponding to the number of detection windows that cover the spectral range. For computational purposes, the spectral phasor transform is expressed as a discretized approximation of the continuous transform as:

$$G = \frac{\sum_c^{N_c} I(c) \cos(2\pi c / N_c)}{\sum_c^{N_c} I(c)}, \tag{8}$$

$$S = \frac{\sum_c^{N_c} I(c) \sin(2\pi c / N_c)}{\sum_c^{N_c} I(c)}, \tag{9}$$

where $I(c)$ is the pixel intensity at channel and $c$ is the total number of channels. Conveniently, even if the total number of spectral acquisition channels is small (in our case 32), the coordinates S and G can be considered quasi-continuous, since the photon counts in each pixel and channel are high enough (~102) to allow a wide range of possible values of the coordinates S and G.

The spectral phasor approach obeys the rules of vector algebra, known as the linear combination of phasors. This property implies that a combination of two independent fluorescent species will appear on the phasor plot at a position that is a linear combination of the phasor positions of the two independent spectral species. The fraction of each component can be determined from the coefficients of the linear combination.

Note that the phase angle $\phi$, and the modulus, $M$ can be obtained through:

$$\phi = \arctan\left(\frac{S}{G}\right) \tag{10}$$

$$M = \sqrt{S^2 + G^2} \tag{11}$$

Data processing and spectral phasor analysis were performed by first converting the Image stacks into.r64 files using the SimFCS software developed at the Laboratory of Fluorescence Dynamics, available on the webpage (https://www.lfd.uci.edu/globals/), and further processed using Python programming code based on the PhasorPy library (https://www.phasorpy.org/).

## Relating contact angles, mechanical parameters, and permittivity contrast

This section is based on the theoretical framework derived in ref. 51. Briefly, the contact angles in Fig. 5a, are related to the force balance between the different surface tensions of the three surface segments pulling along the membrane-condensate contact line. One of these tensions is the interfacial tension of the condensate-buffer interface $\Sigma_{ce}$ (see Fig. 5a). The other two are the mechanical tensions of the two membrane segments in contact with the depleted phase ($\Sigma_{ie}^m$) and condensate ($\Sigma_{ic}^m$), given by ref. 53:

$$\Sigma_{ic}^m = \Sigma + W_{mc} \text{ and } \Sigma_{ie}^m = \Sigma + W_{me} \tag{12}$$

where $\Sigma$ denotes the lateral stress within the membrane, whereas $W_{mc}$ and $W_{me}$ represent the adhesion free energies per unit area of the condensate and of the external depleted phase, relative to the interior solution[53]. The affinity contrast between the condensate and the external buffer is then given by ref. 53:

$$W = \Sigma_{ic}^m - \Sigma_{ie}^m = W_{mc} - W_{me} \tag{13}$$

The geometry-dependent lateral membrane stress $\Sigma$ drops out from the affinity contrast $W$, which is negative if the membrane prefers the condensate phase over the external buffer, and positive otherwise. The force balance in Fig. 5a also implies the geometric relationships[53]

$$\frac{\Sigma_{ie}^m}{\Sigma_{ce}} = \frac{\sin \theta_c}{\sin \theta_i} \text{ and } \frac{\Sigma_{ic}^m}{\Sigma_{ce}} = \frac{\sin \theta_e}{\sin \theta_i} \tag{14}$$

between the surface tensions and the contact angles, as follows from the law of sines for the tension triangle in Fig. 5a. When we introduce Eq. (12) and take the difference of the two equations in Eq. (14), for the affinity contrast $W$ in Eq. (13), we arrive at

$$W = \cos \theta^{in} \Sigma_{ce} \text{ where } \cos \theta^{in} \equiv \frac{\sin \theta_e - \sin \theta_c}{\sin \theta_i} \tag{15}$$

Thus, the rescaled affinity contrast, $W/\Sigma_{ce}$, which is a mechanical quantity related to the adhesion free energies of the membrane segments, is equal to $\cos \theta^{in}$, where the intrinsic contact angle $\theta^{in}$ is a scale-invariant material parameter.

## Reporting summary

Further information on research design is available in the Nature Portfolio Reporting Summary linked to this article.

## Data availability

The source data underlying Figs. 1d–f, 2a–e, 3b, d, e, 4b, d, e, and 5c–e as well as Supplementary Figs. S1a–f, S2a–c, S3a–c, S4a–c, S5a, b, S6, S7, S8a, b, and S9 are provided in an Excel file labelled "Source Data" together with example hyperspectral imaging stacks deposited in the Edmond data repository: https://doi.org/10.17617/3.AB5B8T.

## Code availability

The code generated and used in this study has been deposited in the Edmond repository under accession code https://doi.org/10.17617/3.AB5B8T, and is available under CC BY 4.0 license.

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

## Acknowledgements

A.M. acknowledges support from Alexander von Humboldt Foundation and CONICET. We acknowledge Nannan Chen for providing glycinin, and Clemens Schmitt for the assistance with Raman experiments. We also acknowledge support from the German Academic Exchange Service (Deutscher Akademischer Austauschdienst, DAAD) in the framework of projects 57654674 and 57701619. R.D. acknowledges the ComeInCell network funded by the European Union's Horizon Europe research and innovation program under the Marie Skłodowska-Curie grant agreement No. 101168939. We would like to thank Dr. Leonel Malacrida for the helpful discussions regarding ACDAN hyperspectral imaging and fluorescence mechanisms.

## Author contributions

E.S., A.M. and R.D. conceived the experiments and designed the project. R.D. supervised the project. E.S. performed most of the experiments. E.S. analyzed the data. E.S., A.M. and R.D. wrote the paper.

## Funding

## Competing interests

The authors declare no competing interests.
