## [Transparent Peer Review file · Nature Communications]

Fluorescence-based mapping of condensate dielectric permittivity uncovers hydrophobicity-driven membrane interactions

Corresponding Author: Dr Rumiana Dimova

Version 0:

Reviewer comments:

Reviewer #1

(Remarks to the Author)

What are the noteworthy results?

In this paper, the authors take a conceptual step from considering biomolecular condensates' polarity to permittivity, a parameter which additionally considers the molecules' rotational mobility. They develop a fluorescence microscopy method for quantitative, calibrated read-out of the local permittivity using the fluorophore ACDAN. When applying this method to a variety of condensates consisting of different constituents, they found that permittivity differs widely from each other. As a novel aspect, the researchers explicitly consider and characterize the dilute phase. For the same condensate type, permittivity of both dense and dilute phases varies with macromolecular concentration. Upon wetting of membranes by condensates, the intrinsic contact angle depends on the permittivity of both dense and dilute phase.

Will the work be of significance to the field and related fields?

This work will contribute to the field by providing a new, quantitative fluorescence tool to characterize an important physicochemical parameter of condensates which was up to now very difficult to assess. Moreover, it draws attention to permittivity as a parameter which may be better suited for describing interactions between condensates and membranes than polarity.

How does it compare to the established literature?

The fluorophore ACDAN has been used for years in various contexts, as cited. It's fluorescence emission has previously been analyzed by phasor plots, as cited. What is new is the characterization of ACDAN's emission with respect to permittivity and it's quantitative calibration.

Does the work support the conclusions and claims, or is additional evidence needed? Are there any flaws in the data analysis, interpretation and conclusions?

The authors assume that there are free water molecules inside the condensates whose dipole moment reorientation is sensed by ACDAN. How do they justify this assumption for the different condensate types? Their Raman spectra (S8) show different levels of water relative to peptide.

Moreover, the authors attribute all differences in ACDAN's spectra to different permittivities of the various condensate types and all changes in fluorescence emission of ACDAN upon concentration changes to permittivity changes (Fig. 2). Thus, they assume that there are no changes in macromolecular crowding or ionic strength, which would also influence ACDAN's emission. Yet, there are no control measurements to assess e.g. the level of molecular crowding in the different condensate types or when changing macromolecular concentration, or to assess the ionic strength inside glycinin condensates at increasing NaCl concentrations.

In their experiments, the authors change the initial concentration of the macromolecules (Fig. 3), and show resulting changes in permittivity. For me, the cause for this correlation is not clear: Based on theory, I would expect that varying the total initial

concentration would change the volume fraction of the condensates, but the concentration and crowding inside the condensate should not change. These dense and dilute-phase concentrations are fixed by the phase diagram. Since the permittivity roughly reflects the composition (water fraction, etc.), I would expect permittivity in both dense and dilute phase to remain constant as well. The presented data does not allow me to assess any change in volume fractions.

The observed increase in permittivity of both dense and dilute phases to me signals that the in vitro system is not behaving as an ideal LLPS system, but that the composition of each phase changes, particularly the crowding inside increases or hydration decreases. Or a changing permittivity might reflect ion partitioning changes or non-ideal partitioning of macromolecules. Thus, I would appreciate additional control experiments to monitor macromolecular crowding changes to help interpret these observations.

The authors state that permittivity contrast governs membrane-condensate interactions, i.e. wetting behavior (Fig. 4). Yet, they do not consider other relevant aspects such as the condensate's surface tension, the electrical potential at the condensate interface, or the condensate's viscoelasticity.

The electric potential at the interface of condensates arises from the interplay of multiple factors: molecular polarity and charge distribution, molecular orientation at the surface, permittivity contrast between dilute and dense phase, and ion partitioning. For example, the membrane's surface charge when incorporating charged lipids (DOPS here) would electrostatically interact with the condensate interface electrical potential. I would also expect changing NaCl concentration to have an impact. It would be great if the authors could integrate this into their discussion.

In my understanding, the wetting of membranes by condensates is impacted both by the interfacial electrical potential and the bulk permittivity, in distinct and coupled ways. For highly charged membranes, I would expect the electric potential to dominate the wetting behavior, whereas for weakly charged membranes, I would consider the bulk permittivity to be the dominant factor.

Line 94: "These findings thus establish the permittivity of the coexisting phases as key quantifiable parameters for understanding how biomolecular condensates interact with diverse cellular structures such as membrane-bound organelles and the cytoskeleton" It is not intuitively clear to me why permittivity would dictate condensate-cytoskeleton interactions.

Is the methodology sound?

Yes.

Is there enough detail provided in the methods for the work to be reproduced?

Yes.

Reviewer #2

(Remarks to the Author)

Please see my attached review

Version 1:

Reviewer comments:

Reviewer #1

(Remarks to the Author)

Thank you for the revised manuscript and for addressing my points. I am pleased to see the manuscript has been strengthened by these revisions. The text is now easier to follow and more accessible, and particularly Fig. 1, Fig. 3b and Fig. 4 are easier to understand.

Thanks for providing the additional fluorimetry control measurements of ACDAN emission as a function of NaCl concentration in homogeneous solutions in the new Fig. S7, which clearly exclude an influence of ionic strength / salinity. I appreciate the added discussion point that ACDAN partially binds to hydrophobic pockets within folded BSA, as well as the discussion how a competition between the surface electrical potentials and the bulk permittivity should distinctively affect membrane wetting.

In summary, I recommend the manuscript for publication.

(Remarks on code availability)

Reviewer #2

(Remarks to the Author)

The authors have really done an excellent job in revising this work, both in terms of improving figure clarity, addressing reviewer comments, and improving the manuscript. I remain excited about this work, and really congratulate the authors on a DEEP, very impressive investigation into condensate physical chemistry, the types of which will, I hope, catalyze new ideas/investigations across both biomolecular and synthetic self-assembly systems. I also really appreciate the care and attention the authors gave to both reviewers' comments and questions.

I have two final suggestions that the authors are (truly) free to implement or just ignore, as they want.

During the (long) review process, a paper was preprinted using a range of techniques to explore the internal behaviour of condensates formed from the N-terminal disordered region of DDX4 [Perets et al.: doi:10.1101/2025.11.04.686660]. My reading of both papers is that these are highly complementary; this work uses simple peptides to uncover deep, high-resolution insights into what's going on, with strong explanatory power, while the Perets et al paper uses a battery of techniques to measure a natural protein condensate, but does not offer the same degree of insight into why what's happening. It might be nice to briefly mention this work in the Discussion and perhaps compare/contrast findings/observations. If the authors feel this is an unnecessary distraction, I do not consider this essential.

Second, the response (and changes) in response to the BSA only dielectric result did prompt me to wonder... To what extent is the signal here effectively reporting on a binding affinity for biomolecules in the system? i.e., the assumption is that BSA's direct interaction with hydrophobic pockets on the surface drives deviations from the "bulk" solvent, but to what extent is this interpretation valid for all the data? i.e., deviations from bulk do not reflect a change in the solvent, but simply the partition of the molecule between solvent and a bound state? This is not necessarily an issue, and indeed, perhaps this is functionally how we should be thinking about solvent environments, but nevertheless, I think it might be prudent to raise this question/thought in the discussion. I guess the crux of my thought here is if we interpret the BSA only result based on the specific interactions with hydrophobic patches on BSA... should we not consider that other systems (e.g. other condensate forming proteins that contain folded proteins) may give rise to results that are reporting not on "bulk" condensate permittivity, but on interactions with folded domains in the condensate. I don't believe this is an issue for this work, but I raise this because I think if folks in the future use this approach, we want to avoid a situation where they do not consider this result, and conclude all condensates with folded domains have a MUCH MUCH lower permittivity than those with disordered regions... Just a suggestion to safeguard against future work done by groups with less attention to detail or molecular insight than the authors of this work!

(Remarks on code availability)

We thank the Reviewers for their constructive comments, which helped us improve the clarity of the manuscript. Our responses are provided below in red and are numbered for ease of reference; quoted manuscript text is shown in italics, with new additions in red and unchanged text in black.

Reviewer #1:

What are the noteworthy results?

In this paper, the authors take a conceptual step from considering biomolecular condensates' polarity to permittivity, a parameter which additionally considers the molecules' rotational mobility. They develop a fluorescence microscopy method for quantitative, calibrated read-out of the local permittivity using the fluorophore ACDAN. When applying this method to a variety of condensates consisting of different constituents, they found that permittivity differs widely from each other. As a novel aspect, the researchers explicitly consider and characterize the dilute phase. For the same condensate type, permittivity of both dense and dilute phases varies with macromolecular concentration. Upon wetting of membranes by condensates, the intrinsic contact angle depends on the permittivity of both dense and dilute phase.

Will the work be of significance to the field and related fields?

This work will contribute to the field by providing a new, quantitative fluorescence tool to characterize an important physicochemical parameter of condensates which was up to now very difficult to assess. Moreover, it draws attention to permittivity as a parameter which may be better suited for describing interactions between condensates and membranes than polarity.

How does it compare to the established literature?

The fluorophore ACDAN has been used for years in various contexts, as cited. It's fluorescence emission has previously been analyzed by phasor plots, as cited. What is new is the characterization of ACDAN's emission with respect to permittivity and it's quantitative calibration.

We thank the Reviewer for the positive and encouraging comments and for appreciating the novelty and significance of our work.

Does the work support the conclusions and claims, or is additional evidence needed? Are there any flaws in the data analysis, interpretation and conclusions?

The authors assume that there are free water molecules inside the condensates whose dipole moment reorientation is sensed by ACDAN. How do they justify this assumption for the different condensate types? Their Raman spectra (S8) show different levels of water relative to peptide. Moreover, the authors attribute all differences in ACDAN's spectra to different permittivities of the various condensate types and all changes in fluorescence emission of ACDAN upon concentration changes to permittivity changes (Fig. 2).

A1: We thank the Reviewer for raising this important point and for the opportunity to clarify our interpretation. We would like to emphasize that we do not **assume** the presence of freely reorienting water molecules inside condensates; rather, this conclusion is **deduced** from both theoretical considerations and experimental evidence. We have revised the manuscript to clarify this distinction and to more explicitly justify our interpretation, as follows:

First, we now explicitly acknowledge in the manuscript that a rigorous, quantitative determination of the exact water-to-protein ratio inside condensates is beyond the scope of the present study (for such analyses, we refer to our recent work¹, prepared during the review period). Importantly, Figs. 1 and S1-S3 demonstrate that the redshift of ACDAN's emission spectrum is fully governed by a single physical parameter: the dielectric permittivity of the surrounding medium. This empirical observation is fully consistent with first-principles fluorescence theory, as described by the Lippert–Mataga relation (Eq. 1), which predicts that dipole-dipole interactions between a fluorophore and its dielectric environment produce a systematic shift in emission wavelength. Crucially, this calibration is established not only in aqueous solutions but also across a range of solvents with different dielectric constants, highlighting the generality of the approach. We have revised the text accordingly, now stating:

“Additionally, Fig. 1e,f demonstrates that the redshift of ACDAN's emission spectrum is governed by a single physical quantity, namely the solvent dielectric permittivity. Importantly, this empirical observation is fully consistent with first-principles fluorescence theory, where according to the Lippert–Mataga relation², dipole–dipole interactions between the fluorophore and its dielectric environment give rise to a predictable shift in the emission maximum (see SI). Indeed,

the Lippert–Mataga equation accurately reproduces the calibration data obtained across a broad range of solvents (Fig. 1e,f solid curve), establishing a robust relationship between the maximum emission wavelength λ_{\max} and solvent permittivity ϵ ...

Building on this validated framework, we apply ACDAN hyperspectral imaging to biomolecular condensates (Fig. 2) and find, for example, that K₁₀D₁₀ condensates exhibit a higher dielectric permittivity than R₁₀D₁₀ condensates (note that these two oligopeptides differ by only four atoms between the structures of lysin and arginine). While it is well established that dipole relaxation dynamics in crowded environments is largely dominated by water molecules rather than macromolecular dipoles³⁻⁵, we further substantiate this interpretation through the experiments presented in Fig. S9 (formerly Fig. S8). Specifically, these Raman spectroscopy measurements demonstrate that oligopeptide condensates with higher water molar fractions exhibit higher dielectric permittivity. Similarly, less concentrated PEG-rich and dextran-rich phases in the PEG-dextran ATPS exhibit lower permittivity with decreasing polymer fractions as shown in new Fig. 3b, Fig. S5 and new Table S4. These observations imply that, in such condensates, a significant fraction of water molecules retains sufficient rotational freedom to contribute to dipolar relaxation and thus to the redshift of ACDAN emission. We also refer the reader to our recent preprint¹ for further information of how water-to-protein ratios in biomolecular condensates can be quantified using Raman spectroscopy.

However, as with all fluorescence-based approaches, ACDAN reports the dielectric properties of its immediate molecular environment. The following clarification was added to the introduction:

“As with any solvatochromic probe, ACDAN reports the dielectric properties of its immediate molecular environment.”

While we assume spatial homogeneity at the optical resolution of our measurements, submicroscopic heterogeneities or preferential dye accumulation in specific molecular environments cannot be excluded, as discussed now for the measurements on BSA samples.

“We also observed that BSA-PEG condensates display higher fluorescence intensity, likely reflecting an increased quantum yield of ACDAN and a known tendency of the “sticky” BSA protein to bind small molecules⁶. As a solvatochromic fluorophore, ACDAN probes the dielectric properties in its immediate molecular vicinity rather than a bulk-averaged environment; in this case, the enhanced signal suggests that the measured permittivity predominantly reflects protein-proximal microenvironments rather than a simple bulk average (see Discussion).”

We have also entered a word of caution in the Discussion section, regarding the interpretation of data obtained on “sticky”, highly structured proteins which exhibit hydrophobic pockets – in such systems ACDAN reports the permittivity in the immediate environment of the protein, please see also our answer **A10** to a POINT 3 raised by reviewer 2. The following text was added to the discussion:

“As a note of caution, while water content is a primary determinant of condensate permittivity, additional effects can dominate in systems containing folded proteins with hydrophobic microcavities, as exemplified by BSA. In homogeneous BSA solutions, we measure apparent permittivity values as low as $\epsilon \approx 25\epsilon_0$ at concentrations of ~ 0.05 mM. While reduced water dipolar relaxation in protein hydration layers can partially account for this effect⁷, it cannot explain the nearly order-of-magnitude decrease observed at such low protein volume fractions.

We therefore propose that ACDAN partially binds to hydrophobic pockets within folded BSA, consistent with BSA’s known affinity for small ligands. In such environments, water is highly structured and exhibits an effective permittivity of $\epsilon \approx 5-6\epsilon_0$. Dye molecules bound to these pockets are expected to display enhanced quantum yield⁶ (consistent with the markedly higher ACDAN fluorescence intensity in Fig. 2a), causing protein-associated ACDAN to dominate the fluorescence signal and, respectively, the permittivity readout.

In this regime, ACDAN reports the dielectric properties of its local protein-proximal microenvironment rather than the bulk-averaged permittivity reflecting the water content. Nevertheless, the dielectric differences observed in Figs. 3f-j and S8 are biologically relevant, as enzymatic activity requires a highly hydrated environment capable of sustaining a hydrogen-bonded water network at the protein surface^{8,9}. In this context, the reduced permittivity measured in the condensed phase of PEG-BSA systems aligns with reports of decreased enzymatic activity in the condensed phase of PEG-BSA mixtures compared to homogeneous BSA solutions of similar overall concentrations¹⁰.

We emphasize that local, protein-proximal sensing of dielectric permittivity by ACDAN is specific to folded proteins with well-defined hydrophobic cavities, and is not observed for unstructured polypeptides (K₁₀, R₁₀, D₁₀) or hydrophilic polymers (PDDA, PEG, dextran), where condensate permittivity is governed primarily by the water-to-polymer ratio (Figs. 3b, S5) and is independent of dye concentration (for details see ref. ¹).

Importantly, this observation highlights that dielectric permittivity in condensates and solutions of highly structured proteins can be very heterogeneous at the molecular scale and that local protein-proximal environments may differ

substantially from expectations based on average composition alone. ACDAN-based measurements therefore provide access to the dielectric microenvironments experienced by proteins and small molecules (i.e. environments that are likely most relevant for biochemical function), even when these differ from macroscopic expectations and bulk hydration.”

Thus, they assume that there are no changes in macromolecular crowding or ionic strength, which would also influence ACDAN's emission. Yet, there are no control measurements to assess e.g. the level of molecular crowding in the different condensate types or when changing macromolecular concentration, or to assess the ionic strength inside glycinin condensates at increasing NaCl concentrations.

A2: We thank the Reviewer for raising this important point. We would like to clarify that our analysis does not assume the absence of changes in macromolecular crowding or ionic strength. Rather, these variables are explicitly sampled and incorporated through the calibration step and control experiments that set the basis of our permittivity measurements.

First, a substantial part of the calibration data used in Figs. 1 and S1–S3 is on aqueous polyethylene glycol (PEG) solutions over a wide concentration range. PEG is a well-established macromolecular crowding agent, and these data therefore directly probe the influence of crowding on ACDAN's emission in such systems. In addition, Figs. S6 and S8 present systematic measurements of ACDAN fluorescence as a function of glycinin and BSA concentration, respectively, providing further evidence on the effects of increasing macromolecular density. Together, these datasets demonstrate that changes in macromolecular crowding manifest primarily through predictable shifts in the local dielectric permittivity sensed by ACDAN.

Second, to address the influence of ionic strength, we now include additional fluorimetry measurements of ACDAN emission as a function of NaCl concentration in homogeneous solutions (new Fig. S7, see below). These data show that within the concentration range explored in this study, salinity does not measurably affect ACDAN fluorescence independently of its impact on permittivity. This observation is consistent with previous microscopy-based controls reported in ref. ¹¹, clearly showing that ACDAN fluorescence is independent of salinity changes.

Importantly, variations in macromolecular crowding and ionic strength are not treated as confounding variables in our analysis, but rather as physical mechanisms through which the dielectric properties of the system are modulated. This point is now made explicit in the revised main text, where we discuss how different trajectories through the phase diagram (Fig. 3a–g), lead to systematic and polymer-specific changes in permittivity. These revisions clarify that ACDAN does not report on crowding or ionic strength per se, but on their integrated effect on the local dielectric environment around the dye.

The following text was added:

“Figure 3a-g establishes that changes in macromolecular crowding systematically translate into variations in permittivity sensed by ACDAN across a range of phase-separated systems, with magnitudes that depend strongly on the molecular nature of the macromolecules. In segregative phase separation, increasing PEG or dextran concentrations (Fig. 3a,b), as well as increasing BSA concentration (Fig. 3f,g), enhance macromolecular crowding, leading to higher polymer or protein concentrations in both coexisting phases. These changes are consistently reflected by a reduction in the measured permittivity in each phase.

Similarly, Fig. 3c-e shows that increasing NaCl concentration above the lower critical point of glycinin solutions induces protein condensation, leading to a pronounced increase in macromolecular crowding within the dense phase, which is accompanied by a marked decrease in permittivity. Importantly, Fig. S7 presents spectrofluorimetric controls demonstrating that NaCl concentration alone does not substantially affect ACDAN fluorescence. Consistent microscopy-based controls confirming the salt insensitivity of ACDAN emission have been reported previously¹¹. Together, these results demonstrate that macromolecular crowding and ionic conditions influence ACDAN emission indirectly, through their impact on the local dielectric environment sensed by the probe.”

Figure S7: Control for salt effects on ACDAN-reported permittivity. Spectrofluorimetric measurements of ACDAN fluorescence emission in glycine-free aqueous solutions at increasing NaCl concentrations (legend). No systematic spectral changes are observed. The corresponding permittivity values vary by less than $\sim 4\epsilon_0$ across the explored salt range ($80 \pm 2\epsilon_0$ for 0 mM, $78 \pm 5\epsilon_0$ for 230mM and $82 \pm 2\epsilon_0$ for 460mM NaCl), demonstrating that NaCl alone does not significantly affect the permittivity reported by ACDAN-

In their experiments, the authors change the initial concentration of the macromolecules (Fig. 3), and show resulting changes in permittivity. For me, the cause for this correlation is not clear: Based on theory, I would expect that varying the total initial concentration would change the volume fraction of the condensates, but the concentration and crowding inside the condensate should not change. These dense and dilute-phase concentrations are fixed by the phase diagram.

A3: We thank the reviewer for requesting clarification. In Fig. 3a, as one moves deeper into the two-phase coexistence region (highlighted in green), the total polymer concentration increases (black points), and so do the polymer concentrations within the two coexisting phases. The polymer concentrations in these phases are defined by the endpoints of the tie-lines (dotted lines) that go through the specific studied point in the phase diagram and intersect the binodal (solid line). The reviewer is correct that for measurements performed at polymer concentrations along a single tie-line, the concentrations in the dense and dilute phases remain constant, and only their relative volume fractions change (as described by the lever rule). In our study, however, we explore samples with initial polymer concentrations (black dots) lying on different tie-lines. Consequently, the dense and dilute phases adopt different polymer compositions, as indicated by the crosses on the binodal. The figure caption was slightly modified:

“Figure 3: Hyperspectral imaging of condensate permittivity allows mapping of phase state changes. (a) Sketch of a PEG-dextran phase diagram, see ¹² for experimental values. Crosses (1-2) represent homogeneous PEG-dextran mixtures. Black dots indicate the initial polymer concentration of the tested phase-separated mixtures. The binodal is shown by a black curve and grey dotted lines represent the tie lines. The endpoints of the tie lines indicate the dextran-rich phase compositions (crosses 3-6) and the PEG-rich phase compositions (7-10).”

Since the permittivity roughly reflects the composition (water fraction, etc.), I would expect permittivity in both dense and dilute phase to remain constant as well. The presented data does not allow me to assess any change in volume fractions. The observed increase in permittivity of both dense and dilute phases to me signals that the in vitro system is not behaving as an ideal LLPS system, but that the composition of each phase changes, particularly the crowding inside increases or hydration decreases. Or a changing permittivity might reflect ion partitioning changes or non-ideal partitioning of macromolecules. Thus, I would appreciate additional control experiments to monitor macromolecular crowding changes to help interpret these observations.

A4: As explained above (answer **A3**), because the total polymer concentrations vary for each condition tested (i.e., samples lie on different tie-lines in the phase diagram), the final polymer and water content in both the dense and dilute phases varies, contrary to the assumption of the reviewer. Since ACDAN reports on the local dielectric environment, these changes in composition are directly reflected in the measured permittivity.

The new panel (b) in Fig. 3 (the former panel was moved to Fig. S5a) shows the measured dielectric constant for the homogeneous and phase separated solutions along with the specific compositions of the coexisting phases

now also given in new Table S4. The trends in Figs. 3b and S5 show that overall with increasing polymer concentration, i.e. decreasing water fraction, the dielectric permittivity decreases.

The following material was added:

“Measurements for points (3–10) in Fig. 3a,b were performed by isolating the PEG-rich and dextran-rich phases of the ATPS; corresponding permittivity maps and phase compositions are provided in Fig. S5 and Table S4. Even a small increase in polymer concentration (a few tens of percent), accompanied by a reduction in water fraction, consistently results in lower permittivity, corroborating previous reports¹³. Moreover, Fig. 3a,b shows that changes in total polymer concentration (crosses 1 and 2 for homogeneous and black dots for phase-separated solutions in panel a) result in different levels of macromolecular crowding and water content in both homogeneous and phase-separated regimes, which are quantitatively reflected in the measured permittivity values.”

Figure 3: ... (b) Permittivities of homogeneous and phase separated solutions at different total polymer concentration. The labels next to the data points correspond to the compositions schematically presented in panel (a), see Fig. S5 for hyperspectral maps and Table S4 for exact compositions.

Table S4: Compositions of the PEG-rich and dextran-rich phases in the aqueous two-phase systems presented in Fig. 3a-b in the main text based on the phase diagram presented in ref. ¹⁴. w_p and w_d are the respective total PEG and dextran weight fractions, w_p^P and w_d^P represent the PEG and dextran weight fractions in the PEG-rich phase, w_p^D and w_d^D are the PEG and dextran weight fractions in the dextran-rich phase, and V_P and V_D are the volume fractions occupied by the PEG-rich and the dextran-rich phases within the phase separated solution.

Points, crosses	w_p (%)	w_d (%)	w_p^P (%)	w_d^P (%)	w_p^D (%)	w_d^D (%)	V_P (%)	V_D (%)
1	3.0	4.8	N.A.	N.A.	N.A.	N.A.	N.A.	N.A.
2	3.2	5.1	N.A.	N.A.	N.A.	N.A.	N.A.	N.A.
3, 7	3.6	5.6	5.8	0.8	2.5	8.2	35	65
4, 8	3.8	6.0	6.3	0.6	2.0	9.9	42	58
5, 9	4.1	6.4	6.9	0.3	1.4	12.4	49	51
6, 10	4.3	6.8	7.4	0.2	1.1	13.8	51	49

The authors state that permittivity contrast governs membrane-condensate interactions, i.e. wetting behavior (Fig. 4). Yet, they do not consider other relevant aspects such as the condensate's surface tension, the electrical potential at the condensate interface, or the condensate's viscoelasticity. The electric potential at the interface of condensates arises from the interplay of multiple factors: molecular polarity and charge distribution, molecular orientation at the surface, permittivity contrast between dilute and dense phase, and ion partitioning. For example, the membrane's surface charge when incorporating charged lipids (DOPS here) would electrostatically interact with the condensate interface electrical potential. I would also expect changing NaCl concentration to have an impact. It would be great if the authors could integrate this into their discussion.

A5: We thank the reviewer for these questions. The impact of surface tension is explicitly accounted for in the analytical expression of the intrinsic contact angle through the term Σ_{ce} as we have already specified in the text:

“The force balance between the tensions of the wetted (Σ_{ic}^m) and bare (Σ_{ie}^m) membrane segments and the condensate interfacial tension (Σ_{ce})¹⁵ (see Methods) shapes the vesicle-condensate couple into an axisymmetric system characterized by three apparent contact angles ($\theta_i, \theta_c, \theta_e$) (Fig. 4a and Methods).”

The surface electrical potential of both the condensates and the membrane are indeed expected to influence wetting. As already shown in Fig. 4e, the membrane surface charge impacts the membrane-condensate affinity (compare the measurements for fixed permittivity contrast and altered membrane charge, i.e. different fraction of the charged lipid DOPS, shown by the colored crosses). Presumably, exploring these systems further would yield different intercept of the intrinsic contact angle θ^{in} (and thus the wetting affinity) versus the permittivity contrast $\frac{\epsilon_c - \epsilon_d}{\epsilon_0}$ as tentatively presented in Fig. 4e while the slope may alter as a function of the surface charge of the condensates. This is now further discussed in the text:

“...membrane-condensate interactions emerge from a multifaceted interplay of molecular polarity, surface orientation, charge, and ion partitioning. While previous studies have explored the effect of condensate surface potential, ionic strength¹⁵ and protein sequence¹⁶, our results extend this framework by identifying dielectric permittivity as a quantitative, bulk parameter that directly governs condensate-membrane interactions, revealing an additional and previously unrecognized physical determinant of wetting behavior..

..

The membrane surface charge further modulates this interaction: achieving the same intrinsic contact angle on charged membranes requires a higher permittivity contrast than on neutral membranes (Fig. 4e). Together, these observations indicate that membrane-condensate interactions arise from competing contributions: for highly charged membranes, surface electrostatics dominate and largely mask dielectric effects, whereas for weakly charged membranes, the bulk permittivity contrast between dense and dilute phases becomes the primary determinant of condensate affinity..”

Regarding the viscoelastic properties of condensates, their mechanical response can be decomposed into two distinct contributions. (1) The condensate surface tension provides a restoring force against deformation and coalescence, enforcing a spherical equilibrium shape. This elastic contribution as mentioned above is explicitly captured in our model the term Σ_{ce} . (2) The viscous nature of the condensate bulk is a purely dynamical property that controls the timescale over which the condensate–membrane system relaxes toward equilibrium (see, e.g., Fig. 4a in ref ¹⁵.), without affecting the equilibrium configuration itself. Accordingly, the equilibrium topology of the condensate–membrane system—quantified by the intrinsic contact angle—depends only on interfacial tensions and is independent of condensate viscosity. These considerations have been now emphasized in the main text:

“Notably, while the intrinsic contact angle defining the membrane-condensate equilibrium topology explicitly accounts for the influence of condensate surface tension (see Methods), it is independent from condensate viscosity, which only influences the timescale over which the membrane-condensate system relaxes to its equilibrium configuration.”

In my understanding, the wetting of membranes by condensates is impacted both by the interfacial electrical potential and the bulk permittivity, in distinct and coupled ways. For highly charged membranes, I would expect the electric potential to dominate the wetting behavior, whereas for weakly charged membranes, I would consider the bulk permittivity to be the dominant factor.

A6: We thank the reviewer for this insightful comment. As indicated in our answer to the previous question (**A5**), a couple of paragraphs have been appended to the discussion section detailing how a competition between the surface electrical potentials and the bulk permittivity should distinctively affect the adhesion energy terms that govern membrane-condensate topology in the presented framework.

Line 94: "These findings thus establish the permittivity of the coexisting phases as key quantifiable parameters for understanding how biomolecular condensates interact with diverse cellular structures such as membrane-bound organelles and the cytoskeleton" It is not intuitively clear to me why permittivity would dictate condensate-cytoskeleton interactions.

A7: We thank the reviewer for this comment. As clarified in the revised manuscript, we toned down the sentence, which now reads:

“These findings establish the permittivities of the coexisting phases as key quantifiable parameters for understanding how biomolecular condensates interact with diverse cellular structures such as membrane-bound organelles and – via modulation of surface adhesion energies – potentially other components such as the cytoskeleton. “

We would like to emphasize that the underlying principle is not limited to membranes: we show that permittivities modulate interactions through adhesion energies, which are a general physicochemical characteristic. Thus, for any cellular structure where condensate adhesion or wetting is relevant (such as cytoskeletal filaments or other

structures) variations in the dielectric properties of the condensate could similarly influence affinity and interaction strength. The cytoskeleton is mentioned here as a representative example of a non-membranous structure where such adhesion-mediated effects may occur.

Is the methodology sound?

Yes.

Is there enough detail provided in the methods for the work to be reproduced?

Yes.

Reviewer #2:

This is an exciting paper with several important observations and results. I am supportive of publication, but I have a number of points that I think must be addressed before publication. I want to be clear, I do not see any conclusion that may be arrived at from any of these questions as devaluing the impact of the work and/or reducing my enthusiasm, but I do think there are a number of things the authors should either explicitly rule out or acknowledge as alternative explanations for their observations.

We thank the reviewer for their encouraging and constructive comments. We greatly appreciate the recognition of the importance and impact of our work, and we have carefully addressed all points raised, either by providing additional data, analyses, or clarifications, and by acknowledging alternative explanations where appropriate.

Major points

POINT 1: I would like to see the authors plot condensate permittivity vs. dense-phase concentration—i.e., to what extent is the internal permittivity determined by the concentration of biomolecules within the condensate? The PEG-dextran results in Fig. 3 hint at this, and I recognize obtaining accurate information on condensate dense-phase concentrations is far from trivial, but I do think it would behoove the authors to raise the possibility that condensate permittivity is well-described as being proportional to water content (whereby denser condensates have less water, so have a lower permittivity) and either exclude this possibility (or not, if they cannot directly provide data).

A8: We thank the reviewer for this suggestion. We agree that relating condensate permittivity to water content is an important question. Permittivity data as a function of concentration for the PEG-dextran system were already included in the original submission (Fig. S5), albeit shown only as images and also in Fig. 3. In response to this comment, we have now added a new panel (Fig. 3b – the former panel was moved to Fig. S5a) that explicitly plots the extracted permittivity values for homogeneous solutions as well as for the PEG-rich and dextran-rich phases (the compositions of the individual phases are now also given in new Table S4). The trends in Fig. 3b and S5 show that overall with increasing polymer concentration, i.e. decreasing water fraction, the dielectric permittivity decreases in these systems. The new Fig. 3b and Table S4, and amended text are given in our answer **A4** to reviewer 1.

POINT 2: The authors state: “Importantly, the weak correlation between the type of interaction driving LLPS and condensate permittivity (Fig. 2e) suggests that condensate dielectric properties cannot be predicted solely based on the chemical nature of components or the LLPS mechanism” - but Fig. 2e shows no such correlation and, moreover, I feel like looking at the components here these trends are not necessarily so unexpected (although extremely cool to see!), at least to me. As such, I think if the authors are to make such a claim, they need to formally quantify what they mean by this, or remove such a claim. For example, the K10D10 vs. R10D10 is actually what at least I would have expected (which is very exciting!) based on a large body of exact work exploring the differences between lysine and arginine in phase separation (e.g. see [1-2] for specific investigation into hydration, but more broadly there is a tremendous body of work exploring the K/R difference in phase separation; the fact Arg and Lys give very different behaviors despite varying only by four amino acids is well-established across many papers).

A9: We thank the reviewer for raising this point and have reformulated the statement in the text:

“Additionally, the weak correlation between the type of interaction driving LLPS and condensate permittivity (Fig. 2e) suggests that condensate dielectric properties cannot be predicted solely based on the type of LLPS mechanism. A striking example is the comparison between K_{10} - D_{10} and R_{10} - D_{10} condensates both formed via similar electrostatic heterotypic interactions; despite only four-atom difference between lysin and arginine, their permittivities differ drastically, i.e. by $\sim 20\epsilon_0$ (Fig. 2e). These observations are consistent with in silico assessments of differential hydration energies in arginine- versus lysin-rich assemblies^{17, 18}.”

POINT 3: Maybe I'm misunderstanding this, but it seems that figure 3g suggest that in a homogenous solution of 50 μM BSA the solution permittivity is ~ 25 (SEE FIGURE TO THE RIGHT)? This seems... impossibly low? Is this right? Does this make sense? If protein solutions of tens of μM show such drastic changes in permittivity, is the story here not that actually the condensate interiors are shocking similar to their exterior, given the massive difference in protein concentration?

A10: We thank the reviewer for raising this very important and initially puzzling observation. We agree that a permittivity value of $\epsilon \approx 25$ for a homogeneous solution containing only tens of μM BSA is surprisingly low, and we were similarly concerned when first observing this result. We have therefore expanded both the Results and Discussion sections to explicitly address this point.

Our analysis indicates that BSA exemplifies a different case among the systems studied. Unlike the small polypeptides (K_{10} , R_{10} , D_{10}) or hydrophilic polymers (PDDA, PEG, dextran), BSA is a folded, ligand-binding protein known to contain hydrophobic cavities. We observe that BSA-containing condensates exhibit a markedly higher ACDAN fluorescence intensity (Fig. 2a), despite identical dye concentrations, consistent with an increased quantum yield upon dye binding, as previously reported for a related (although markedly different) DAN probe⁶. In this context, ACDAN does not predominantly report a volume-averaged permittivity of the bulk solution, but rather the dielectric properties of the immediate protein microenvironment. Protein hydration is known to strongly suppress water dipolar relaxation within the first hydration layers, yielding locally reduced effective permittivity⁷. However, simple hydration-layer effects alone cannot account for the nearly order-of-magnitude decrease observed at very low BSA concentrations, where proteins and hydration layers together occupy only a few percent of the volume.

We therefore propose that potential binding of ACDAN to hydrophobic pockets in BSA leads to (i) a strong increase of its quantum yield in low-permittivity environments and (ii) associated enhancement of the detected signal from these regions, biasing the reported permittivity toward values in the protein proximity. We now explicitly state that, in this regime, ACDAN reports the local permittivity in the immediate vicinity of the protein rather than the bulk (with high water content).

We have investigated the question further, and in a study that we conducted after the initial submission and during the long review period (ref. ¹), we confirm that this behavior is not observed for unstructured polypeptides or hydrophilic polymers lacking hydrophobic microcavities, where the measured permittivity is governed by water-to-polymer content (the new Fig. 3b and Fig. S5 demonstrate this for the PEG-dextran system); below we include an adapted figure from our new study¹.

Figure R1: Similar water content assessed with Raman microscopy in two different condensate types based on BSA (which exhibit hydrophobic cavities) and PDDA (unstructured) show very different permittivity values indicated above the bars.

Thus, while BSA measurements highlight an aspect (or limitation) that we now clearly acknowledge, this does not undermine the general applicability of the method to condensates formed by intrinsically disordered proteins, peptides, or synthetic polymers. More generally, these observations highlight an important conceptual distinction between water content and local hydrophobicity in biomolecular condensates. While the volume fraction of water is often used as a proxy for the physicochemical environment within condensates, our results show that folded proteins such as BSA can create substructural microenvironments whose dielectric properties differ markedly from those predicted by bulk composition alone. In this sense, ACDAN does not merely report the average hydrophobicity and water content, but rather the dielectric microenvironment experienced by small molecules or functional groups in the immediate vicinity of proteins. This distinction is particularly relevant for biological processes (such as enzyme activity, ligand binding, or small-molecule partitioning) that depend on local polarity rather than on bulk hydration. We conclude that water content and local hydrophobicity are not equivalent observables in heterogeneous, structured condensates. We have clarified this point in the text.

Text added after Fig 2:

“We also observed that BSA-PEG condensates display higher fluorescence intensity, likely reflecting an increased quantum yield of ACDAN and a known tendency of the “sticky” BSA protein to bind small molecules⁶. As a solvatochromic fluorophore, ACDAN probes the dielectric properties in its immediate molecular vicinity rather than a bulk-averaged environment; in this case, the enhanced signal suggests that the measured permittivity predominantly reflects protein-proximal microenvironments rather than a simple bulk average (see Discussion).”

Text added in the Discussion:

“As a note of caution, while water content is a primary determinant of condensate permittivity, additional effects can dominate in systems containing folded proteins with hydrophobic microcavities, as exemplified by BSA. In homogeneous BSA solutions, we measure apparent permittivity values as low as $\epsilon \approx 25\epsilon_0$ at concentrations of ~ 0.05 mM. While reduced water dipolar relaxation in protein hydration layers can partially account for this effect⁷, it cannot explain the nearly order-of-magnitude decrease observed at such low protein volume fractions.

We therefore propose that ACDAN partially binds to hydrophobic pockets within folded BSA, consistent with BSA’s known affinity for small ligands. In such environments, water is highly structured and exhibits an effective permittivity of $\epsilon \approx 5-6\epsilon_0$. Dye molecules bound to these pockets are expected to display enhanced quantum yield⁶ (consistent with the markedly higher ACDAN fluorescence intensity in Fig. 2a), causing protein-associated ACDAN to dominate the fluorescence signal and, respectively, the permittivity readout.

In this regime, ACDAN reports the dielectric properties of its local protein-proximal microenvironment rather than the bulk-averaged permittivity reflecting the water content. Nevertheless, the dielectric differences observed in Figs. 3f-j and S8 are biologically relevant, as enzymatic activity requires a highly hydrated environment capable of sustaining a hydrogen-bonded water network at the protein surface^{8,9}. In this context, the reduced permittivity measured in the condensed phase of PEG-BSA systems aligns with reports of decreased enzymatic activity in the condensed phase of PEG-BSA mixtures compared to homogeneous BSA solutions of similar overall concentrations¹⁰.

We emphasize that local, protein-proximal sensing of dielectric permittivity by ACDAN is specific to folded proteins with well-defined hydrophobic cavities, and is not observed for unstructured polypeptides (K_{10} , R_{10} , D_{10}) or hydrophilic

polymers (PDDA, PEG, dextran), where condensate permittivity is governed primarily by the water-to-polymer ratio (Figs. 3b, S5) and is independent of dye concentration (for details see ref. ¹).

Importantly, this observation highlights that dielectric permittivity in condensates and solutions of highly structured proteins can be very heterogeneous at the molecular scale and that local protein-proximal environments may differ substantially from expectations based on average composition alone. ACDAN-based measurements therefore provide access to the dielectric microenvironments experienced by proteins and small molecules (i.e. environments that are likely most relevant for biochemical function), even when these differ from macroscopic expectations and bulk hydration.

...

Beyond these fundamental insights, our method provides a sensitive dielectric descriptor of coexisting phases across different regions of a phase diagram (Fig. 3). While it does not yield a direct readout of phase composition, it robustly captures relative changes in the physicochemical environments of dense and dilute phases, allowing the quantification of subtle changes induced by hydrotropes like ATP (Fig. 3h,i). Considering ATP's central role in cellular physiology, our results suggest that ATP not only influences protein solubility^{19, 20} but also acts as a key modulator of condensate interactions with other cellular components, by affecting permittivity contrast between coexisting phases."

POINT 4: The authors (I think) claim that one can use PEG-free BSA solutions to calibrate BSA concentration in the PEG-BSA condensates (Line 298; "Given that BSA is the main hydrophobic component of the mixture, this representation allows a qualitative estimate of BSA concentration in different regions of the phase-separated system"). However, they have just shown that varying PEG concentration can have a huge impact on permittivity (Fig. 3a/b - points 7/8/9/10 retain similar dextra concentrations but vary PEG, showing BIG changes in permittivity), and of course both BSA and PEG concentrations change concomitantly as you linearly increase BSA concentration, so to me at least, this logic doesn't hold at all? Further, the statement "Given that BSA is the main hydrophobic component of the mixture" is a strong assertion which as far as I can see is unsupported by data and, honestly seems counterintuitive (I'd expect BSA to be less hydrophobic than PEG).

A11: We thank the reviewer for this important clarification. We fully agree that permittivity measurements cannot be used to quantitatively infer BSA concentration in PEG-BSA condensates, since PEG and BSA concentrations vary concomitantly and PEG itself has a strong and independent influence on permittivity, as clearly demonstrated in Fig. 3a,b. In the revised manuscript, we have therefore removed the discussion suggesting that ACDAN-based permittivity measurements allow estimation of protein or polymer concentrations in coexisting phases. Instead, we now emphasize that permittivity reports on the local dielectric and hydration environment sensed by the probe, which integrates contributions from water content, macromolecular crowding, and specific probe-environment interactions, but does not uniquely map onto composition. We believe this revised framing is more accurate and strengthens the conceptual clarity of the work:

"Permittivity measurements in PEG-BSA phase-separated systems further illustrate how the dielectric microenvironment differs between condensed and depleted phases. In these mixtures, both protein and polymer concentrations vary simultaneously across coexisting phases, and PEG itself strongly influences the permittivity (Fig. 3a,b). Thus, permittivity values as sensed by ACDAN reflect the combined effects of water content, macromolecular crowding, but, importantly, also local probe-microenvironment interactions as we discuss below.

...

Overall, these findings demonstrate that permittivity measurements provide a sensitive readout of changes in dielectric microenvironment within coexisting phases, enabling qualitative tracking of how phase behavior responds to molecular perturbations such as ATP, without relying on assumptions about phase composition."

POINT 5: In Fig 4b, I think it would be important to in-line in the figure show how permittivity changes as a function of NaCl in the absence of glycinin - i.e. the presumption here is that the change in permivity is due to a salt-dependent change in dilute-phase glycinin concentration (i.e. as salt increases, glycerin is released from condensates decreasing the concentration in the dilute phase and reducing permittivity - which also speaks to the question about tie lines in Fig. 3c). HOWEVER, while I'd agree with that interpretation, I think it would important to show this is NOT a weird salt effect!

A12: We thank the reviewer for raising this point. In the absence of condensates, NaCl concentration alone does not substantially alter the permittivity reported by ACDAN. We now explicitly demonstrate this by including new control data in Fig. S7 (included in our answer **A2** to reviewer 1), which shows spectrofluorimetric measurements

of ACDAN fluorescence in homogeneous aqueous solutions over the range of NaCl concentrations used in Fig. 4b. These measurements confirm that the emission spectrum, and thus the inferred permittivity, remains essentially unchanged with increasing salt concentration.

Accordingly, the salt-dependent permittivity changes observed in Fig. 4b arise from salt-induced changes in phase composition, rather than from a direct interaction between NaCl and the dye. In particular, increasing ionic strength shifts the phase equilibrium, altering the glycinin concentration in the dilute phase and thereby modulating its permittivity.

To make this explicit, we have added the following text to the manuscript:

“Importantly, Fig. S7 presents spectrofluorimetric controls demonstrating that NaCl concentration alone does not substantially affect ACDAN fluorescence. Consistent microscopy-based controls confirming the salt insensitivity of ACDAN emission have been reported previously¹¹. Together, these results demonstrate that macromolecular crowding and ionic conditions influence ACDAN emission indirectly, through their impact on the local dielectric environment sensed by the probe”

POINT 6: Contact angle is, of course, going to depend massively on the size of condensates vs. membrane, and I honestly couldn't actually work out how these experiments were being done. Are the “membranes” here GUVs? Is there a normalization factor to consider the GUV/condensate radius ratio? I think we need a lot more detail to understand how the results in 4c were generated.

A13: We thank the reviewer for calling for clarifications on this matter. Yes, the membranes used in these experiments are GUVs. We have now expanded the relevant paragraph in the main text to explicitly describe how contact angles are defined, measured, and interpreted in the vesicle–condensate geometry:

“The force balance between the tensions of the wetted (Σ_{ic}^m) and bare (Σ_{ie}^m) membrane segments and the condensate interfacial tension (Σ_{ce})^{15, 21} (see Methods) shapes the vesicle-condensate couple into an axisymmetric system characterized by three apparent contact angles ($\theta_i, \theta_c, \theta_e$) (Fig. 4a and Methods). These apparent contact angles are extracted from confocal cross-sections passing through the symmetry axis of the vesicle–condensate system. While they fully describe the observed geometry, their values depend on extrinsic parameters such as condensate and vesicle size, as well as the degree of vesicle deflation, which vary between individual vesicle–condensate pairs in a sample. By contrast, at the nanometer scale, the membrane is smoothly curved at the three-phase contact line (black circle in Fig. 4a)²² and wetting is characterized by the intrinsic contact angle θ_{\square}^{in} ²³. This intrinsic angle is a material parameter that is independent of system geometry or size¹⁵ and quantifies the membrane preferential affinity for the protein-rich (condensed) versus protein-poor (depleted) phase¹⁵.”

Importantly, no normalization with respect to the vesicle-to-condensate size ratio is required, because the quantity reported in Fig. 4c is the intrinsic contact angle θ_{\square}^{in} , not the apparent angles measured directly from the images. While the apparent contact angles indeed vary strongly between different vesicle-condensate pairs, the intrinsic contact angle is invariant for a given membrane-condensate system and therefore constitutes a true material property.

This distinction between apparent and intrinsic contact angles and the resulting geometry independence of θ_{\square}^{in} , has been discussed in detail in ref.¹⁵ and demonstrated experimentally both for protein condensates and aqueous two-phase systems²³. In particular, ref.¹⁵ presents examples of vesicle–condensate pairs within the same sample that exhibit markedly different apparent geometries (apparent contact angles as assessed from the images), yet yield identical intrinsic contact angles. This is illustrated in Fig. R2 (adapted from Supplementary Fig. S12 of ref.¹⁵), which compares two DOPC GUVs interacting with K₁₀-D₁₀ condensates of very different relative sizes. Despite large differences in the apparent contact angles, both systems yield the same intrinsic contact angle ($\approx 55^\circ$), demonstrating that θ_{\square}^{in} is independent of the vesicle-to-condensate size ratio.

Figure R2: Differences in vesicle and condensate sizes (see images and left axis of graph) do not affect the intrinsic contact angle (diamonds, right axis). DOPC vesicles (magenta) in contact with K_{10} -D $_{10}$ condensates (yellow). In image 1, the condensate is larger than the vesicle; in image 2, the opposite geometry applies. Despite the distinct apparent contact angles, the intrinsic contact angle is identical in both cases ($\approx 55^\circ$).

POINT 7: Figure 4c is, I think, the key result for this section, but in my view, totally uninterpretable in isolation. Firstly, the authors should provide a visual schematic of what varying the contact angle means. Second, we don't know what the actual control parameters here are; percentages are randomly listed, new symbols (L_d and L_o), and no images show condensate membrane interaction. I'd strongly encourage the authors to expand this figure into multiple additional panels, each conveying a single key idea clearly and accessibly. This result is – I think – really interesting, but as it stands, I don't think most readers will get it!

A14: We thank the reviewer for this constructive and important comment. We agree that the original version of Fig. 4c was difficult to interpret in isolation, and we have revised this figure to improve clarity, accessibility, and physical intuition. First, we expanded the figure to include schematic illustrations that define condensate–membrane wetting and explicitly show how changes in the intrinsic contact angle correspond to different affinity regimes (new panel b). These schematics provide a direct geometrical interpretation of contact angles and their physical meaning, addressing the reviewer's request for visual guidance. Second, we clarified the experimental control parameters by explicitly separating how permittivity contrast is tuned, either via polymer concentration (PEG-dextran) or ionic strength (glycinin-NaCl), and by showing how these variables affect the dense and dilute phases independently (new panel d). Third, we reorganized the data presentation so that each panel conveys a single key idea: (i) the physical definition of wetting and intrinsic contact angle, (ii) how permittivity contrast emerges from phase behavior, and (iii) how this contrast quantitatively correlates with membrane affinity across multiple systems and membrane compositions (panel e). The wetting measurements in panel (e) rely on previously published datasets, and we now explicitly indicate their origin, membrane composition (Liquid-ordered vs. liquid-disordered, charged vs. neutral), and the corresponding permittivity contrasts derived from our measurements. The revised figure and caption read:

Figure 4: Membrane wetting by condensates is modulated by the permittivity contrast between coexisting phases. (a) Schematic illustration of a giant unilamellar vesicle (yellow) wetted by a condensate (blue, permittivity ϵ_c) and surrounded by the polymer-depleted phase (light green, permittivity ϵ_d). The apparent contact angles facing the vesicle interior (θ_i), the external phase (θ_e), and the condensate (θ_c), define the system equilibrium morphology. These angles, measured from confocal cross-sections of giant vesicles in contact with condensates, reflect a balance of interfacial forces at the three-phase contact line (black circle): the condensate interfacial tension Σ_{ce} , and the mechanical tensions of the two membrane segments in contact with the depleted condensate-free phase (Σ_{ie}^m , orange) and condensate phase (Σ_{ic}^m , purple). At the nanometer scale, the membrane *does not exhibit a kink but* is smoothly curved (inset), and wetting is described by the intrinsic contact angle $\theta_{\square}^{\text{in}} = \arccos\left(\frac{\sin \theta_e - \sin \theta_c}{\sin \theta_i}\right)$ ¹⁵, see Methods. (b) Schematic illustrations of condensate wetting flat lipid membranes (top) and vesicles (bottom) illustrating distinct affinity regimes quantified by the intrinsic contact angle $\theta_{\square}^{\text{in}}$ defined in (a) and characteristic morphologies. Increasing polymer or salt concentration enhances the membrane affinity of dextran and glycine condensates^{12, 15}. (c) Hyperspectral permittivity maps of glycine condensates at increasing NaCl concentrations. Top: dielectric maps of fields of view ($36.9 \times 36.9 \mu\text{m}^2$). The color bar represents the logarithmic permittivity scale. Bottom: boxplot showing that increasing salt primarily lowers the permittivity of the condensate-free (depleted) phase, while the condensate phase remains relatively unaffected. (d) Permittivity contrasts ($\epsilon_c - \epsilon_d$) between dense (condensate) and external (depleted) phases as a function of polymer concentration (PEG-dextran, top, data from Figs. 3b and S5), and ionic strength (glycine-NaCl, bottom, data from panel (c)). (e) The intrinsic contact angle $\theta_{\square}^{\text{in}}$, describing condensate-membrane affinity, increases linearly with the permittivity contrast ($\epsilon_c - \epsilon_d$), indicating that stronger permittivity differences enhance membrane wetting by the condensate as illustrated schematically on the left. All experimental datasets were fitted with a linear function (dashed lines). Crosses display $\theta_{\square}^{\text{in}}$ data from ref.¹⁵ for glycine condensates interacting with DOPC:DOPS membranes at various NaCl concentrations (pure DOPC, orange), and for 100 mM NaCl but different charged lipid content (10, 20 and 40 mol% DOPS, respectively shown in rose, vermilion and red); the respective permittivity contrast $\epsilon_c - \epsilon_d$ is from panel (c). Dotted lines passing through data points for DOPS-doped membranes have the same slope as the charge-free membrane and serve as guides to the eye. Circles show $\theta_{\square}^{\text{in}}$ data from ref.¹² for PEG-dextran ATPS droplets at different polymer concentrations in contact with either liquid-disordered (DOPC:DPPC:Chol, 64:15:21, blue) or liquid-ordered (DOPC:DPPC:Chol, 13:44:43, green) membranes. The corresponding permittivity contrasts were interpolated from data presented in Fig. 3b.

Note that throughout the manuscript we have changed the notation for the permittivity of the depleted phase from ϵ_e to ϵ_d . This modification avoids ambiguity in vesicle-condensate systems formed by PEG-dextran ATPS, where phase separation can occur not only outside but also inside giant vesicles and both the condensate and

depleted phases (dextran-rich and PEG-rich, respectively) reside within the vesicle interior rather than in an external environment.

POINT 8: The crux of the result in Figure 4, as I understand it (I think —I found this result confusing, so I may be misunderstanding!) is that the condensate's internal permittivity dictates the contact angle with membranes. But if permittivity scales with the driving force for phase separation (which to my mind is at least consistent with the data being shown? If that interpretation is wrong, it'd be an important thing to highlight) then how can the authors exclude the alternative (and perhaps simpler?) hypothesis that contact angle is simply proportional to the difference in free energy of demixing between dense and dilute phases, which (1) determines the permittivity and (2) determines the contact angle? I think a fairly direct test for this would be to take PEG:Dextran condensates where at different total concentrations of both (where clearly the driving forces don't change as volume fraction of PEG and dextran are increased - i.e. diagonal increase as shown in Fig 3a).

A15: We thank the reviewer for raising this important conceptual point and for the opportunity to clarify our interpretation.

The central result of Fig. 4 is that membrane affinity is governed **not by the permittivity of the condensate**, but rather **by the permittivity contrast between the dense and depleted phases ($\epsilon_c - \epsilon_d$)**. This contrast emerges from phase behavior but is not equivalent to the free energy of demixing.

If the intrinsic contact angle were controlled primarily by the demixing free energy, one would expect it to peak at the condition of maximal thermodynamic driving force for phase separation: in the glyciniin–NaCl system, this corresponds to an intermediate salt concentration (~100 mM), where the distance to both upper and lower binodals is maximal (Fig. 3c). Instead, the intrinsic contact angle increases monotonically with NaCl concentration¹⁵. This monotonic behavior mirrors the evolution of the permittivity contrast: while the permittivity of the dense phase remains essentially constant with salt, the permittivity of the dilute phase decreases continuously, leading to an increasing $\epsilon_c - \epsilon_d$ (Fig. 4c,d). These trends are therefore inconsistent with a mechanism governed solely by demixing free energy, but are naturally explained by a dielectric mechanism that depends on phase-specific permittivities.

Importantly, this conclusion is further supported by PEG–dextran systems explored along trajectories that cross different tie lines (Fig. 3a), where the demixing mechanism remains unchanged while the permittivity contrast, and correspondingly the wetting affinity, varies (Fig. 4e). We have clarified these points in the manuscript by adding the following paragraph:

“It is important to distinguish changes in condensate-membrane affinity from variations in the free energy of demixing. If these quantities were directly coupled, one would expect the intrinsic contact angle in the glyciniin-membrane system to exhibit a maximum at the point of largest separation from both the upper and lower binodals in Fig. 3c, where the thermodynamic driving force for phase separation is maximal. Instead, the intrinsic contact angle increases monotonically with NaCl concentration¹⁵. This behavior is consistent with the observed evolution of the permittivity contrast: while the permittivity of the dense phase remains largely insensitive to salt, that of the dilute phase decreases continuously (Fig. 4c). These trends indicate that membrane affinity is governed by phase-specific dielectric properties rather than by the demixing free energy alone. Additional contributions may arise from NaCl-dependent changes in protein conformation, as suggested previously¹¹.”

POINT 9: As written, I worry the presentation may be a little too technical for a broad audience. I think this can be

A16: We thank the reviewer for this important comment. We agree that some technical concepts (particularly spectral phasor analysis) may not be immediately intuitive to a broad readership. To improve accessibility, we have simplified Fig. 1 and added a new schematic (in panel c) that visually explains how spectral shifts are encoded in phasor space:

In addition, we revised the main text to more explicitly separate the physical principle (dipolar relaxation and emission redshift) from the analytical implementation (Gaussian fitting or phasor analysis). The revised text now emphasizes that the key observable is the redshift of AC DAN fluorescence, while the phasor representation is introduced as a fit-free and robust computational tool that yields equivalent results and can be skipped by readers not interested in methodological details. We believe these changes improve the readability of the manuscript without compromising technical rigor.

“To translate this solvent-dependent spectral redshift into a quantitative measure of dielectric permittivity (i.e. build a calibration curve), we characterized AC DAN’s spectral response in a broad panel of homogeneous solvents with known permittivity, using both hyperspectral microscopy (Fig. 1c,d) and spectrofluorimetry (see SI). While both approaches yield equivalent permittivity calibrations for homogeneous samples (Figs. S1–S3), spectrofluorimetry is limited to bulk (cuvette) measurements and lacks spatial resolution, limiting its ability to resolve distinct dielectric properties in phase-separated (or heterogeneous) samples. We therefore employ hyperspectral microscopy, which records a full emission spectrum at each image pixel, generating spatially resolved spectral maps (Fig. 1c,d). This allows local permittivity variations to be visualized directly within heterogeneous systems, including phase-separated biomolecular condensates. To construct a calibration curve, we measured homogeneous reference solutions of known permittivity (Fig. 1d) and analyzed the spectra using two approaches: Gaussian fitting and a fit-free Fourier-space analysis based on spectral phasors²⁴. Both methods yield equivalent results for simple spectra, with the phasor approach offering additional robustness and extensibility for complex spectral profiles.”

Minor points

I think an important thing the authors need to explicitly state (with data or references to support), OR acknowledge as a possible limitation, is whether anything other than dielectric permittivity can influence the emission spectra. Specifically, the assumption here, based on the calibration curve, is that there is a bijective mapping between emission and dielectric permittivity (i.e., the only determinant of emission spectra is dielectric

permittivity). This may be true and well established, but at the very least, the authors should (1) state this assumption explicitly, ideally with references/data to support the assumption as fact, or (2) state this assumption and acknowledge that assumption as a potential limitation. For example, if ACDAN's molecules clustered more or less as a function of their environment, could this alter spectral properties beyond them acting as a passive probe? If they found themselves arranged in a particular orientation, might this change their apparent fluorescence (e.g., as ThT or Congo red change upon amyloid interaction or EthBr on DNA intercalation)? Neither of these is true, but at the same time, it would give me confidence if the authors could explicitly comment on the presumed 1:1 mapping. FWIW, I think the whole point of the phasor plot is to show that quenching can be directly accounted for in the analysis - however, I'm not sure such a strong conclusion can actually be drawn (i.e., I don't know!). Also, to be explicit, I do not think raising this weakens the work at all!

A17: We thank the reviewer for this thoughtful and constructive comment. We agree that it is very important to explicitly state the underlying assumption that links ACDAN emission spectra to dielectric permittivity and to clarify the conditions under which this assumption holds.

Throughout the manuscript, we interpret ACDAN fluorescence as reporting the dielectric permittivity of the dye's immediate microenvironment. Our calibration in simple solvents, together with validation in polymer solutions and in condensates formed by unstructured polypeptides, demonstrates a bijective mapping between emission maximum and permittivity over a broad range of chemistries, dye concentrations, and measurement modalities (microscopy and spectrofluorimetry). In these systems, ACDAN behaves as a passive solvatochromic probe, and no evidence for dye clustering, orientation effects, or specific binding is observed. We stated this in a paragraph following Fig. 1:

“The close agreement between our calibration and validation datasets (Fig. 1e,f) demonstrates robust sensitivity that is consistent across different solvent chemistries and independent of the measurement platform (microscopy or spectrofluorimetry). In these systems, ACDAN behaves as a passive solvatochromic probe, allowing us to treat the dye emission redshift, quantified by λ_{max} , as a bijective function of the permittivity of the dye immediate microenvironment: each λ_{max} corresponds to a unique permittivity ϵ , and vice versa. While we expect this relationship to hold broadly, potential limitations in systems containing folded proteins with specific binding sites are addressed in the discussion section.”

At the same time, we fully agree that this bijection is not necessarily universal across other systems. We therefore explicitly discuss folded proteins with hydrophobic cavities (e.g. BSA) as a distinct regime, where ACDAN can partially bind to protein microenvironments and report local dielectric properties that do not correspond to the bulk water content. In such cases, the measured permittivity reflects the protein-proximal environment sensed by ACDAN rather than yielding a simple volume-averaged dielectric constant. This limitation is now clearly discussed in the Discussion. The included new paragraphs are given in our answer **A10** above.

Importantly, rather than undermining the approach, this distinction highlights that ACDAN-based measurements are sensitive to molecular-scale dielectric heterogeneities, which may be particularly relevant for understanding biochemical function within condensates.

Given the broad readership of Nature Communications, Figure 1 is likely impenetrable for most non-specialist readers. Which is not to say it's unclear or poor, but simply too specialized. I'd encourage the authors to consider reworking this and showing it to non-spectroscopy colleagues to gauge clarity as to what is being shown. For example, I would not assume the general reader knows what information a phasor plot provides. The authors write: “The phasor plot analysis involves computing the discretized Fourier transform of the dye emission spectrum and representing it as a point in a two-dimensional 'phasor plot” (Fig. 1b, bottom right panel)” - which tells us “what it is” (perhaps) but offers little insight into “why it is”? What question is this analysis allowing us to answer? To be clear, this critique is meant to help expand the manuscript's interpretability to a broader audience

A18: We thank the reviewer for these suggestions. Figure 1 was simplified and expanded with an additional schematic panel that introduces the phasor plot at an intuitive level (see answer A16 for the updated figure). The revised figure now explicitly illustrates why the phasor representation is useful: it provides a compact, fit-free way to map each fluorescence spectrum (i.e., each pixel) onto a two-dimensional space, enabling direct visual comparison of local dielectric environments across heterogeneous samples. This representation allows the reader

to immediately see how spectral shifts (and therefore local permittivity) vary spatially, without requiring familiarity with Fourier analysis.

We also revised the text describing the phasor plot to clarify the question it answers, namely: how to transform spatially resolved fluorescence spectra into a robust, quantitative metric of local dielectric permittivity that is independent of intensity, dye concentration, or photobleaching.

The following clarifying paragraphs were included:

“In summary, the phasor plot analysis provides a fit-free representation of fluorescence spectra, allowing each pixel to be mapped onto a two-dimensional space where spectral shifts, rather than absolute intensity, can be directly visualized and compared. In this context, the phasor position serves as a compact descriptor of the local dielectric environment sensed by ACDAN.”

and

“While both approaches yield equivalent results, this comparison serves several purposes: it verifies that our permittivity measurements are robust across analytical frameworks and instrumental platforms; it demonstrates that the canonical Gaussian fit, which encodes only the peak emission wavelength, can be matched by a fit-free, two-dimensional spectral phasor approach capturing both spectral redshift (phase) and broadening (modulus, reflecting quenching and spectral quality); and it shows that the phasor method can be applied to more complex spectra where simple Gaussian fitting is insufficient, enabling straightforward permittivity mapping in challenging systems¹¹.”

The authors refer to “the Lippert-Mataga equation” - again, this is nicely described in terms of what, but for a general readership, this paragraph (line 136 onwards) would benefit tremendously from introducing WHY - why is this analysis important.

A19: We thank the reviewer for the suggestion. We revised the paragraph introducing the Lippert–Mataga equation to motivate why it is relevant. We now explicitly state that Fig. 1e,f demonstrate an empirical one-to-one relationship between the ACDAN emission redshift and solvent permittivity, and that the Lippert–Mataga relation provides the physical foundation explaining this observation. In essence, this framework shows that the spectral shift of ACDAN is governed by a single physical quantity - the dielectric constant, which justifies our use of the emission maximum as a quantitative reporter of permittivity.

“Additionally, Fig. 1e,f demonstrates that the redshift of ACDAN’s emission spectrum is governed by a single physical quantity, namely the solvent dielectric permittivity. Importantly, this empirical observation is fully consistent with first-principles fluorescence theory, where according to the Lippert–Mataga relation², dipole–dipole interactions between the fluorophore and its dielectric environment give rise to a predictable shift in the emission maximum (see SI). Indeed, the Lippert–Mataga equation accurately reproduces the calibration data obtained across a broad range of solvents (Fig. 1e,f solid curve), establishing a robust relationship between the maximum emission wavelength λ_{\max} and solvent permittivity ϵ ...”

The authors move into Fig 3 (PEG-dextran condensates) without at all explaining this new system, and I’d encourage the authors to FIRST introduce this system and why they are using it before reporting on the results in Fig. 3. It’s also not apparent (from the figure) once phase separation happens if condensates are PEG-rich or dextran-rich, so this would be something to probably explicitly show. Finally, there is presumably also water present, meaning this is really a ternary system? Not an issue, just not immediately clear to me from the figure.

A20: We thank the reviewer for raising these points. In the revised manuscript, we now explicitly introduce the PEG-dextran aqueous two-phase system (ATPS) before discussing the data in Fig. 3 – we explain why this system was chosen, how phase separation occurs, and which polymer constitutes the dense phase (please see our answer **A4** to reviewer 1):

“As a benchmark system, we first examine a canonical aqueous two-phase system (ATPS) composed of PEG and dextran. At low polymer concentrations, they form a homogeneous aqueous solution, while increasing the concentration of either polymer induces LLPS into a PEG-rich phase and a dextran-rich dense phase (Fig. 3a). This PEG-dextran ATPS is a well-established model system with robustly characterized phase diagrams (see e.g. ref. ¹⁴), making it ideally suited to validate our approach and to systematically track how dielectric permittivity varies with macromolecular composition.

Although water is present in both phases (making this formally a ternary system) the PEG-dextran phase diagram and tie lines provide a way to evaluate the water content and macromolecular crowding in each phase based on changes in polymer concentration. Building on this knowledge, Figure 3b illustrates how variations in the initial PEG and dextran concentrations, schematically indicated as points in the phase diagram (Fig. 3a), translate into distinct permittivity values.”

In Fig. 3c, do the authors have a sense of where the tie-lines should fall? This would be nice to have to then relate the results in 3e to 3c.

A21: We unfortunately do not have information about the tie lines for the glycinin-NaCl phase diagram reported in ref.²⁵.

In Fig 3f and 3h, I think it would be handy and important to have even just schematics of the expected phase behavior as a function of BSA/PEG (3f) and K10 -D10+ATP (3h), recognizing that the ternary phase diagram is more complex to show. I'd also encourage the authors to be explicit about why the ATP addition has the effect it does (i.e., acting to weaken the interactions that underlie condensate formation and, as such, reducing the concentration of components (or I suppose increasing water content?) inside the condensate).

A22: We thank the reviewer for raising this point, we added an illustrative panel h to Figure 3 to illustrate the expected phase behavior associated with the addition of ATP and included a discussion in the text. We thank the reviewer for this helpful suggestion. In the revised manuscript, we have added schematic phase diagrams (new panel Fig. 3h) to qualitatively illustrate the expected phase behavior of the PEG–BSA and K₁₀–D₁₀ systems, and explicitly described the role of ATP in modulating phase behavior:

Figure 3: ... (h) Schematic phase diagrams illustrating the expected phase behavior of the PEG–BSA system (top) and the K₁₀-D₁₀ system (bottom). The solid black curves denote the binodals, and the black dotted lines represent tie lines through the initial overall composition (black dot). Black crosses indicate the compositions of the coexisting dense and depleted phases in the absence of ATP. The purple dotted curves illustrate the expected shift of the binodal upon ATP addition, reflecting weakened intermolecular interactions. Correspondingly, purple crosses indicate the altered compositions of the coexisting phases in the presence of ATP.

The following text was added:

“Figure 3h schematically summarizes the expected effects of ATP addition on the phase behavior of the investigated systems. ATP acts to weaken the intermolecular interactions that drive phase separation, resulting in a lower macromolecule concentration (and correspondingly higher water content) within the dense phase, alongside an increased protein/polymer concentration in the depleted phase. In the phase diagram representation, this manifests as a shift of the binodal boundaries away from the origin upon ATP addition, consistent with the observed reduction in permittivity contrast between coexisting phases as shown in Fig. 3i,j.”

In Figure 4, please identify the components in the figure.

A23: The components have been identified in the new version of the figure and the caption.

References

1. Sabri, E., Mangiarotti, A., Schmitt, C. & Dimova, R. Label-free in situ approach for characterizing the macromolecular composition and water content in biomolecular condensates. *bioRxiv*, 2025.2010.2020.683555 (2025).
2. Lakowicz, J. Principles of fluorescence spectroscopy. *University of Maryland School of Medicine Baltimore* **132** (2006).

3. Thoke, H.S., Thorsteinsson, S., Stock, R.P., Bagatolli, L.A. & Olsen, L.F. The dynamics of intracellular water constrains glycolytic oscillations in *Saccharomyces cerevisiae*. *Scientific Reports* **7**, 16250 (2017).
4. Thoke, H.S. et al. Tight coupling of metabolic oscillations and intracellular water dynamics in *Saccharomyces cerevisiae*. *PLoS One* **10**, e0117308 (2015).
5. Begarani, F. et al. Capturing metabolism-dependent solvent dynamics in the lumen of a trafficking lysosome. *ACS nano* **13**, 1670-1682 (2019).
6. Fernandes, A.J.F.C. et al. 4-Dimethylamino-beta-nitrostyrene, a fluorescent solvatochromic probe to estimate the apparent dielectric constant in serum albumin: Experimental and molecular dynamics studies. *J. Photochem. Photobiol. A: Chem.* **433**, 114197 (2022).
7. Ghosh, R., Banerjee, S., Hazra, M., Roy, S. & Bagchi, B. Sensitivity of polarization fluctuations to the nature of protein-water interactions: Study of biological water in four different protein-water systems. *The Journal of Chemical Physics* **141** (2014).
8. Smolin, N., Oleinikova, A., Brovchenko, I., Geiger, A. & Winter, R. Properties of spanning water networks at protein surfaces. *The Journal of Physical Chemistry B* **109**, 10995-11005 (2005).
9. Nakasako, M. Water-protein interactions from high-resolution protein crystallography. *Philosophical Transactions of the Royal Society of London. Series B: Biological Sciences* **359**, 1191-1206 (2004).
10. Lamy, H. et al. Kinetic Study of the Esterase-like Activity of Albumin following Condensation by Macromolecular Crowding. *Biomacromolecules* **25**, 2803-2813 (2024).
11. Mangiarotti, A. et al. Biomolecular condensates modulate membrane lipid packing and hydration. *Nature Commun.* **14**, 6081 (2023).
12. Liu, Y., Agudo-Canalejo, J., Grafmüller, A., Dimova, R. & Lipowsky, R. Patterns of Flexible Nanotubes Formed by Liquid-Ordered and Liquid-Disordered Membranes. *ACS Nano* **10**, 463-474 (2016).
13. Sato, T., Niwa, H., Chiba, A. & Nozaki, R. Dynamical structure of oligo (ethylene glycol) s-water solutions studied by time domain reflectometry. *The Journal of chemical physics* **108**, 4138-4147 (1998).
14. Liu, Y., Lipowsky, R. & Dimova, R. Concentration Dependence of the Interfacial Tension for Aqueous Two-Phase Polymer Solutions of Dextran and Polyethylene Glycol. *Langmuir* **28**, 3831-3839 (2012).
15. Mangiarotti, A., Chen, N., Zhao, Z., Lipowsky, R. & Dimova, R. Wetting and complex remodeling of membranes by biomolecular condensates. *Nature Commun.* **14**, 2809 (2023).
16. Holland, J., Nott, T.J. & Aarts, D.G.A.L. Intrinsic hydrophobicity of IDP-based biomolecular condensates drives their partial drying on membrane surfaces. *The Journal of Chemical Physics* **162** (2025).
17. Fossat, M.J., Zeng, X. & Pappu, R.V. Uncovering differences in hydration free energies and structures for model compound mimics of charged side chains of amino acids. *The Journal of Physical Chemistry B* **125**, 4148-4161 (2021).
18. Armentia, L., López, X. & De Sancho, D. Arginine versus Lysine: Molecular Determinants of Cation- π Interactions in Biomolecular Condensates. *bioRxiv*, 2025.2010. 2001.679751 (2025).
19. Patel, A. et al. ATP as a biological hydrotrope. *Science* **356**, 753-756 (2017).
20. Bagatolli, L.A. To see or not to see: Lateral organization of biological membranes and fluorescence microscopy. *Biochim. Biophys. Acta-Biomembr.* **1758**, 1541-1556 (2006).
21. Lipowsky, R. Response of membranes and vesicles to capillary forces arising from aqueous two-phase systems and water-in-water droplets. *The Journal of Physical Chemistry B* **122**, 3572-3586 (2018).
22. Zhao, Z. et al. Super-Resolution Imaging of Highly Curved Membrane Structures in Giant Vesicles Encapsulating Molecular Condensates. *Adv. Mater.* **34**, 2106633 (2022).
23. Kusumaatmaja, H., Li, Y., Dimova, R. & Lipowsky, R. Intrinsic Contact Angle of Aqueous Phases at Membranes and Vesicles. *Phys. Rev. Lett.* **103**, 238103 (2009).
24. Vorontsova, I. et al. In vivo macromolecular crowding is differentially modulated by aquaporin 0 in zebrafish lens: Insights from a nanoenvironment sensor and spectral imaging. *Science Advances* **8**, eabj4833 (2022).
25. Chen, N., Zhao, Z., Wang, Y. & Dimova, R. Resolving the Mechanisms of Soy Glycinin Self-Coacervation and Hollow-Condensate Formation. *ACS Macro Letters* **9**, 1844-1852 (2020).

Reponses to the Reviewers' comments

Reviewer #1:

Thank you for the revised manuscript and for addressing my points. I am pleased to see the manuscript has been strengthened by these revisions. The text is now easier to follow and more accessible, and particularly Fig. 1, Fig. 3b and Fig. 4 are easier to understand.

Thanks for providing the additional fluorimetry control measurements of ACDAN emission as a function of NaCl concentration in homogeneous solutions in the new Fig. S7, which clearly exclude an influence of ionic strength / salinity.

I appreciate the added discussion point that ACDAN partially binds to hydrophobic pockets within folded BSA, as well as the discussion how a competition between the surface electrical potentials and the bulk permittivity should distinctively affect membrane wetting.

In summary, I recommend the manuscript for publication.

We thank the reviewer for their careful reading and encouraging comments. We are pleased that the revisions improved the clarity and accessibility of the manuscript.

Reviewer #2:

The authors have really done an excellent job in revising this work, both in terms of improving figure clarity, addressing reviewer comments, and improving the manuscript. I remain excited about this work, and really congratulate the authors on a DEEP, very impressive investigation into condensate physical chemistry, the types of which will, I hope, catalyze new ideas/investigations across both biomolecular and synthetic self-assembly systems. I also really appreciate the care and attention the authors gave to both reviewers' comments and questions.

We thank the reviewer for their encouraging and constructive comments.

I have two final suggestions that the authors are (truly) free to implement or just ignore, as they want. During the (long) review process, a paper was preprinted using a range of techniques to explore the internal behaviour of condensates formed from the N-terminal disordered region of DDX4 [Perets et al.: doi:10.1101/2025.11.04.686660]. My reading of both papers is that these are highly complementary; this work uses simple peptides to uncover deep, high-resolution insights into what's going on, with strong explanatory power, while the Perets et al paper uses a battery of techniques to measure a natural protein condensate, but does not offer the same degree of insight into why what's happening. It might be nice to briefly mention this work in the Discussion and perhaps compare/contrast findings/observations. If the authors feel this is an unnecessary distraction, I do not consider this essential.

We thank the reviewer for this suggestion. We agree that the work by Perets et al. is complementary in exploring the internal behavior of natural protein condensates. In the revised discussion, we have added a brief mention of this work in the Discussion:

“Complementary preprint work has explored the internal dynamics of natural protein condensates¹, highlighting the broader relevance of condensate microenvironments.”

Second, the response (and changes) in response to the BSA only dielectric result did prompt me to wonder... To what extent is the signal here effectively reporting on a binding affinity for biomolecules in the system? i.e., the assumption is that BSA's direct interaction with hydrophobic pockets on the surface drives deviations from the "bulk" solvent, but to what extent is this interpretation valid for all the data? i.e., deviations from bulk do not reflect a change in the solvent, but simply the partition of the molecule between solvent and a bound state? This is not necessarily an issue, and indeed, perhaps this is functionally how we should be thinking about solvent environments, but nevertheless, I think it might be prudent to raise this question/thought in the discussion. I guess the crux of my thought here is if we interpret the BSA only result based on the specific interactions with hydrophobic patches on BSA... should we not consider that other systems (e.g. other condensate forming proteins that contain folded proteins) may give rise to results that are reporting not on "bulk" condensate permittivity, but on interactions with folded domains in the condensate. I don't believe this is an issue for this work, but I raise this

because I think if folks in the future use this approach, we want to avoid a situation where they do not consider this result, and conclude all condensates with folded domains have a MUCH MUCH lower permittivity than those with disordered regions... Just a suggestion to safeguard against future work done by groups with less attention to detail or molecular insight than the authors of this work!

We appreciate this insightful point. In the Discussion, we have added a short paragraph to highlight that in condensates containing folded proteins, ACDAN signals may reflect interactions with hydrophobic pockets or structured domains rather than the bulk solvent permittivity, and in the abstract reiterated that the dye reports the dielectric permittivity in its immediate molecular environment. The new paragraph reads:

We note that in condensates containing folded proteins, ACDAN signals may reflect local interactions with hydrophobic pockets rather than bulk solvent permittivity, and future applications of this method should consider that low measured permittivity may arise from structured protein-proximal environments rather than reduced water content.

Reference

1. Perets, E.A. et al. Direct Optical Quantification of Chain Collapse, Reduced Dielectric, and Water Release Driving Protein Phase Separation. *bioRxiv*, 2025.2011.2004.686660 (2025).

This is an exciting paper with several important observations and results. I am supportive of publication, but I have a number of points that I think must be addressed before publication. I want to be clear, I do not see any conclusion that may be arrived at from any of these questions as devaluing the impact of the work and/or reducing my enthusiasm, but I do think there are a number of things the authors should either explicitly rule out or acknowledge as alternative explanations for their observations.

Major points

POINT 1: I would like to see the authors plot condensate permittivity vs. dense-phase concentration—i.e., to what extent is the internal permittivity determined by the concentration of biomolecules within the condensate? The PEG-dextran results in Fig. 3 hint at this, and I recognize obtaining accurate information on condensate dense-phase concentrations is far from trivial, but I do think it would behoove the authors to raise the possibility that condensate permittivity is well-described as being proportional to water content (whereby denser condensates have less water, so have a lower permittivity) and either exclude this possibility (or not, if they cannot directly provide data).

POINT 2: The authors state: “Importantly, the weak correlation between the type of interaction driving LLPS and condensate permittivity (Fig. 2e) suggests that condensate dielectric properties cannot be predicted solely based on the chemical nature of components or the LLPS mechanism” - but Fig. 2e shows no such correlation and, moreover, I feel like looking at the components here these trends are not necessarily so unexpected (although extremely cool to see!), at least to me. As such, I think if the authors are to make such a claim, they need to formally quantify what they mean by this, or remove such a claim. For example, the $K_{10}D_{10}$ vs. $R_{10}D_{10}$ is actually what at least I would have expected (which is very exciting!) based on a large body of exact work exploring the differences between lysine and arginine in phase separation (e.g. see [1-2] for specific investigation into hydration, but more broadly there is a tremendous body of work exploring the K/R difference in phase separation; the fact Arg and Lys give very different behaviors despite varying only by four amino acids is well-established across many papers).

POINT 3: Maybe I'm misunderstanding this, but it seems that figure 3g suggest that in a homogenous solution of 50 μ M BSA the solution permittivity is ~ 25 (SEE FIGURE TO THE RIGHT)? This seems... impossibly low? Is this right? Does this make sense? If protein solutions of tens of μ M show such drastic changes in permittivity, is the story here not that actually the condensate interiors are shocking *similar* to their exterior, given the massive difference in protein concentration?

POINT 4: The authors (I think) claim that one can use PEG-free BSA solutions to calibrate BSA concentration in the PEG-BSA condensates (Line 298; “Given that BSA is the main hydrophobic

component of the mixture, this representation allows a qualitative estimate of BSA concentration in different regions of the phase-separated system”). However, they have just shown that varying PEG concentration can have a huge impact on permittivity (Fig. 3a/b - points 7/8/9/10 retain similar dextra concentrations but vary PEG, showing BIG changes in permittivity), and of course both BSA and PEG concentrations change concomitantly as you linearly increase BSA concentration, so to me at least, this logic doesn't hold at all? Further, the statement “Given that BSA is the main hydrophobic component of the mixture” is a strong assertion which as far as I can see is unsupported by data and, honestly seems counterintuitive (I'd expect BSA to be less hydrophobic than PEG) .

POINT 5: In Fig 4b, I think it would be important to in-line in the figure show how permittivity changes as a function of NaCl in the absence of glycinin - i.e. the presumption here is that the change in permivity is due to a salt-dependent change in dilute-phase glycinin concentration (i.e. as salt increases, glycerin is released from condensates decreasing the concentration in the dilute phase and reducing permittivity - which also speaks to the question about tie lines in Fig. 3c). HOWEVER, while I'd agree with that interpretation, I think it would be important to show this is NOT a weird salt effect!

POINT 6: Contact angle is, of course, going to depend massively on the size of condensates vs. membrane, and I honestly couldn't actually work out how these experiments were being done. Are the “membranes” here GUVs? Is there a normalization factor to consider the GUV/condensate radius ratio? I think we need a lot more detail to understand how the results in 4c were generated.

POINT 7: Figure 4c is, I think, the key result for this section, but in my view, totally uninterpretable in isolation. Firstly, the authors should provide a visual schematic of what varying the contact angle means. Second, we don't know what the actual control parameters here are; percentages are randomly listed, new symbols (L_d and L_o), and no images show condensate membrane interaction. I'd strongly encourage the authors to expand this figure into multiple additional panels, each conveying a single key idea clearly and accessibly. This result is – I think – really interesting, but as it stands, I don't think most readers will get it!

POINT 8: The crux of the result in Figure 4, as I understand it (I think —I found this result confusing, so I may be misunderstanding!) is that the condensate's internal permittivity dictates the contact angle with membranes. But if permivity scales with the driving force for phase separation (which to my mind is at least consistent with the data being shown? If that interpretation is wrong, it'd be an important thing to highlight) then how can the authors exclude the alternative (and perhaps simpler?) hypothesis that contact angle is simply proportional to the difference in free energy of demixing between dense and dilute phases, which (1) determines the permittivity and (2) determines the contact angle? I think a fairly direct test for this would be to take PEG:Dextran condensates where at different total concentrations of both (where clearly the driving forces don't change as volume fraction of PEG and dextran are increased - i.e. diagonal increase as shown in Fig 3a).

POINT 9: As written, I worry the presentation may be a *little* too technical for a broad audience. I think this can be e

Minor points

I think an important thing the authors need to explicitly state (with data or references to support), OR acknowledge as a possible limitation, is whether anything other than dielectric permittivity can influence the emission spectra. Specifically, the assumption here, based on the calibration curve, is that there is a bijective mapping between emission and dielectric permittivity (i.e., the only determinant of mission spectra is dielectric permittivity). This may be true and well established, but at the very least, the authors should (1) state this assumption explicitly, ideally with references/data to support the assumption as fact, or (2) state this assumption and acknowledge that assumption as a potential limitation. For example, if ACDAN's molecules clustered more or less as a function of their environment, could this alter spectral properties beyond them acting as a passive probe? If they found themselves arranged in a particular orientation, might this change their apparent fluorescence (e.g., as ThT or Congo red change upon amyloid interaction or EthBr on DNA intercalation)? Neither of these is true, but at the same time, it would give me confidence if the authors could explicitly comment on the presumed 1:1 mapping. FWIW, I *think* the whole point of the phasor plot is to show that quenching can be directly accounted for in the analysis - however, I'm not sure such a strong conclusion can actually be drawn (i.e., I don't know!). Also, to be explicit, I do not think raising this weakens the work at all!

Given the broad readership of *Nature Communications*, Figure 1 is likely impenetrable for most non-specialist readers. Which is not to say it's unclear or poor, but simply too specialized. I'd encourage the authors to consider reworking this and showing it to non-spectroscopy colleagues to gauge clarity as to what is being shown. For example, I would not assume the general reader knows what information a phasor plot provides. The authors write: "The phasor plot analysis involves computing the discretized Fourier transform of the dye emission spectrum and representing it as a point in a two-dimensional 'phasor plot' (Fig. 1b, bottom right panel)" - which tells us "*what it is*" (perhaps) but offers little insight into "*why it is*"? What question is this analysis allowing us to answer? To be clear, this critique is meant to help expand the manuscript's interpretability to a broader audience

The authors refer to "the Lippert-Mataga equation" - again, this is nicely described in terms of what, but for a general readership, this paragraph (line 136 onwards) would benefit tremendously from introducing WHY - why is this analysis important.

The authors move into Fig 3 (PEG-dextran condensates) without at all explaining this new system, and I'd encourage the authors to FIRST introduce this system and why they are using it before reporting on the results in Fig. 3. It's also not apparent (from the figure) once phase separation happens if condensates are PEG-rich or dextran-rich, so this would be something to probably explicitly show. Finally, there is presumably also water present, meaning this is really a ternary system? Not an issue, just not immediately clear to me from the figure.

In Fig. 3c, do the authors have a sense of where the tie-lines should fall? This would be nice to have to then relate the results in 3e to 3c.

In Fig 3f and 3h, I think it would be handy and important to have even just schematics of the expected phase behavior as a function of BSA/PEG (3f) and K_{10} -D₁₀+ATP (3h), recognizing that the ternary phase diagram is more complex to show. I'd also encourage the authors to be explicit about why the ATP addition has the effect it does (i.e., acting to weaken the interactions that underlie condensate formation and, as such, reducing the concentration of components (or I suppose increasing water content?) inside the condensate).

In Figure 4, please identify the components in the figure.

References

[1] Fossat, M. J., Zeng, X. & Pappu, R. V. Uncovering differences in hydration free energies and structures for model compound mimics of charged side chains of amino acids. *J. Phys. Chem. B* **125**, 4148–4161 (2021).

[2] Armentia, L., Lopez, X. & De Sancho, D. Arginine versus lysine: Molecular determinants of cation- π interactions in biomolecular condensates. *bioRxiv* 2025.10.01.679751 (2025). doi:10.1101/2025.10.01.679751